# Morphological and Genetics Support for a Hitherto Undescribed Spotted Cat Species (Genus *Leopardus*; Felidae, Carnivora) from the Southern Colombian Andes

**DOI:** 10.3390/genes14061266

**Published:** 2023-06-15

**Authors:** Manuel Ruiz-García, Myreya Pinedo-Castro, Joseph Mark Shostell

**Affiliations:** 1Laboratorio de Genética de Poblaciones Molecular-Biología Evolutiva, Departamento de Biología, Facultad de Ciencias, Pontificia Universidad Javeriana, Cra 7A, No 43-82, Bogotá 110231, Colombia; mozpinedo@gmail.com; 2Math, Science and Technology Department, University of Minnesota Crookston, 2900 University Ave., Crookston, MN 56716, USA; joseph.shostell@gmail.com

**Keywords:** Colombia, *Leopardus narinensis*, mitogenomes, Nariño cat, new neotropical wild cat, nuclear markers

## Abstract

In 1989, a skin of a small spotted cat, from the Galeras Volcano in southern Colombia (Nariño Department), was donated to the Instituto Alexander von Humboldt (identification, ID 5857) at Villa de Leyva (Boyacá Department, Colombia). Although originally classified as *Leopardus tigrinus*, its distinctiveness merits a new taxonomic designation. The skin is distinct from all known *L. tigrinus* holotypes as well as from other *Leopardus* species. Analysis of the complete mitochondrial genomes from 44 felid specimens (including 18 *L. tigrinus* and all the current known species of the genus *Leopardus*), the mt*ND5* gene from 84 felid specimens (including 30 *L. tigrinus* and all the species of the genus *Leopardus*), and six nuclear DNA microsatellites (113 felid specimens of all the current known species of the genus *Leopardus*) indicate that this specimen does not belong to any previously recognized *Leopardus* taxon. The mt*ND5* gene suggests this new lineage (the Nariño cat as we name it) is a sister taxon of *Leopardus colocola*. The mitogenomic and nuclear DNA microsatellite analyses suggest that this new lineage is the sister taxon to a clade formed by Central American and trans-Andean *L. tigrinus* + (*Leopardus geoffroyi* + *Leopardus guigna*). The temporal split between the ancestor of this new possible species and the most recent ancestor within *Leopardus* was dated to 1.2–1.9 million years ago. We consider that this new unique lineage is a new species, and we propose the scientific name *Leopardus narinensis*.

## 1. Introduction

Many new species of neotropical mammals have been recently described. Some of these species may be artifacts based on insufficient molecular, ethological, ecological, and karyological data to determine reproductive isolation from other similar taxa as well as extreme typological applications of the Phylogenetic Species Concept (PSC) [1,2]. This may be the case for the 12 alleged “new species” of gracile capuchins (*Cebus*) [3], or the cases of *Mazama bricenii* [4], *Nasuella meridensis* [5], *Odocoileus lasiotis* [6,7], *Pecari maximus* [8], *Inia araguaiaensis* [9], and *Tapirus kabomani* [10,11,12]. On the other hand, there is strong evidence to support the recognition of other new mammal species in the Neotropics, for example, *Bradypus pygmaeus* [13], *Bassarycion neblina* [14], *Mico humilis* [15], *Mico munduruku* [16], *Mico schneideri* [17], *Myotis attenboroughi* [18], *Mindomys kutuku* [19], and different species of *Cyclopes* (for instance, *Cyclopes rufus*, *Cyclopes thomasi*, etc.) (Miranda et al., 2018) [20] to cite a few.

Currently, 11 felid species are identified in Latin America. Eight of them form a monophyletic group named the ocelot lineage [21,22,23,24,25,26,27,28,29,30,31,32,33]. The ocelot lineage includes the ocelot (*Leopardus pardalis*), the margay (*Leopardus wiedii*), the Andean cat (*Leopardus jacobita*), the Pampas or colocolo cat (*L. colocola*; however, see [34]), the kodkod (*L. guigna*), the Geoffroy’s cat (*L. geoffroyi*), the oncilla or tigrina (*L. tigrinus*), and the recently differentiated southern tigrina (*Leopardus guttulus*—[30,31,32]).

The taxonomy of the tigrina has been confusing for centuries. Schreber [35,36] first used the scientific name *Felis tigrina* and illustrated this species with a plate named “Le Margay” [37]. It was based on a specimen from Cayenne, French Guiana. Thus, biologists have long been confused about what is a tigrina and what is a margay. Gray [38] described a supposed new species, *Felis pardinoides*, with “India” as its type locality, but later changed the type locality to Bogota, Colombia [39]. In the same time period, *Felis guttula* was described, and southern Brazil was listed as its type locality [40]. Additional tigrina taxa were described at the beginning of the 20th century, including *Felis pardinoides oncilla* (Costa Rica), *Felis pardinoides andina* (Ecuador), *Felis carrikeri* (Costa Rica), *Felis pardinoides emerita* (Venezuela), and *Felis emiliae* (Ceara State, Brazil). Allen [41] used the genus *Margay* for the tigrina and defined two additional tigrina taxa: *Margay tigrina elenae* (Antioquia Department, Colombia) and *Margay caucensis* (Cauca Department, Colombia). The traditional taxonomy that was used until recently was proposed by Cabrera [42,43] and followed by Wozencraft (in Wilson and Reeder [44]). It comprised four subspecies: *F. t. oncilla* (including *F. p. oncilla* and *F. carrikeri*), *F. t. pardinoides* (including *F. pardinoides*, *F. p. andina*, *F. p. emerita*, *M. t. elenae*, and *M. caucensis*), *F. t. tigrina* (including *F. tigrina* and *F. emiliae*), and *F. t. guttula* (including *F. guttula*). However, the last subspecies was recently claimed as a new species (*L. guttula*) based on genetic data [30,31,32]. The distribution of the other tigrina subspecies are as follows: *L. t. oncilla* inhabits Panama, Costa Rica as far north as Nicaragua. *L. t. pardinoides* inhabits the Andean region including western Venezuela, Colombia, Ecuador, Peru, Bolivia, and northwestern Argentina. *L. t. tigrinus* lives in eastern Venezuela, the Guianas, and northeastern Brazil. 

Pelage characteristics and morphometrics from a large collection of *L. tigrinus* support the existence of three morphogroups [45]. Morphogroup I contains specimens from Central America, as well as specimens from northern, northwestern, and western South America (Costa Rica, Colombia, Venezuela, Ecuador, Peru, Guyana, Suriname, and northwestern Argentina). The color of these tigrinas is brown and ranges from orangish-brown to yellowish-brown. These specimens have a greyish-brown underfur, a white or light grey venter, and medium-sized rosettes that form oblique bands. The bands are arranged in a scapular-inguinal direction on the sides of the body. Following Nascimento and Feijó [45], this taxon is *L. tigrinus* in “senso stricto”. However, large mitochondrial genetic heterogeneity exists among the Central American and Andean tigrina populations that would be included in this morphogroup [46]. Thus, the morphology [45] and the mitochondrial genetics do not align [46]. Morphogroup II contains specimens from the northeastern and central Brazil area. Their color ranges from a light yellowish-brown to pale yellow or pale greyish-buff. They have small rosettes that rarely form oblique bands. The rosettes have thin and discontinuous black rims. This morphogroup was considered a new species, *Leopardus emiliae* [45]. Morphogroup III contains specimens from southern Brazil, Paraguay, and northeastern Argentina and corresponds to *L. guttulus*. They have a dark yellowish-brown ground color, which is lighter on the sides of the body. They also have a white or light grey venter and small rosettes on the sides of the body. Thus, based on this study [45], the number of *Leopardus* species should be increased to nine (with the addition of *L. emiliae*).

During a molecular study to resolve these complex relationships among different populations of *L. tigrinus* [46], we found a felid skin from the Nariño Department in southern Andean Colombia. This skin was dried in the sun with no chemical tanning or other preservation. This was important because the quality of DNA obtained from this skin was high. No other part of the animal (skull, bones, etc.) was available. We rapidly observed that its phenotype differs from the morphological traits of other tigrina taxa as well as from other species of the *Leopardus* genus (for instance, a strong reddish coloration). Although the reddish coloration may be due to an erythristic mutation, we decided to molecularly analyze the skin of this specimen. Therefore, we used the entire mitogenome, and the mitochondrial (mt) *ND5* gene to compare this specimen with all the *Leopardus* species recognized today. Additionally, six nuclear microsatellites were genotyped for this specimen and another 112 felid specimens (including all species of the *Leopardus* genus, although some taxa within *L. tigrinus* and *L. colocola* were not included in this analysis). The molecular comparisons suggest that this felid skin could represent a new and previously undetected lineage, taxon, or even species within the *Leopardus* genus.

## 2. Materials and Methods

### 2.1. Samples

From the unknown felid species, a unique skin specimen (no skull nor bones were obtained) was available from the mammalian collection of the Instituto von Humboldt (ID 5857) at Villa de Leyva (Boyaca Department; Colombia). The specimen was collected in 1989 at the Galeras Volcano, Nariño Department in southern Colombia. Hereafter this sample is referred to as the Nariño cat.

Following the extraction of DNA from this skin (3 cm^2^), we sequenced the complete mitogenome to compare it with a dataset of other *Leopardus* specimens. We also sequenced the mt*ND5* gene to compare it with a bigger dataset of other *Leopardus* specimens (including in both cases all the species recognized of the *Leopardus* genus) and genotyped six nuclear microsatellites, which were also compared with the genetic profiles of all the recognized *Leopardus* species.

The first mitochondrial dataset contained complete mitochondrial genomes. We sequenced complete mitochondrial genomes of 32 specimens. Additionally, we also retrieved full mitogenome sequences from GenBank (12 specimens, for a total of 44 specimens representing nine species; Appendix A). The second mitochondrial dataset contained sequences of the mt*ND5* gene from 84 specimens and nine species (Appendix A). We selected the mt*ND5* gene because the highest number of sequences are available for it in GenBank. All newly sequenced mitogenomes including the one from the Nariño cat were submitted to GenBank (accession numbers: MG230196.1-MG230251.1).

Additionally, we genotyped a total of 113 wild cat specimens (comprising nine species plus the Nariño cat) at six microsatellite loci. These samples were: (1) The Nariño cat sample, (2) Two samples of *L. geoffroyi* from Bolivia and Paraguay, (3) One sample of *L. guigna* from Chile, (4) Eight samples of Andean tigrina, *L. t. pardinoides*, six from Colombia, one from Venezuela, and one from Peru, (5) 13 samples of *L. wiedii*, four from Colombia, four from Peru, two from Guatemala, two from Bolivia, and one from Venezuela, (6) 14 samples of *L. pardalis*, eight from Colombia, three from Ecuador, one from Brazil, one from Peru, and one from Bolivia, (7) 20 samples of *L. jacobita*, 12 from Peru and eight from Bolivia, (8) 39 samples of *L. colocola* (13 *L. c. garleppi*, 12 *L. c. steinbachi*, six *L. c. budini*, and eight *L. c. cruscinus*) from Peru, Bolivia, and Argentina, (9) Six samples of *Felis catus* from Spain, (10) Nine samples of *Herpailurus yagouaroundi*, two from Colombia, two from Peru, two from Venezuela, one from Guatemala, one from French Guiana, and one from Brazil.

We also analyzed samples of *F. catus* and *H. yagouaroundi* for both mitochondrial and nuclear markers because these cat species live in the southern Colombian Andes and could potentially hybridize with *Leopardus t. pardinoides*, *L. wiedii*, *L. pardalis*, and *L. colocola* that inhabit this geographical area.

### 2.2. Mitochondrial DNA

We extracted and isolated DNA from hair, skin, and muscle samples using the QIAamp DNA Micro Kit (Qiagen, Inc.) for the mitogenomic analysis (a total of 16,756 base pairs, bp, including the fragment of mt*ND5* gene, 315 bp in length). DNA extraction from muscle fibers followed the protocol “DNA Purification from Tissues”. For some skin samples, optimized DNA extraction procedures had to be employed [47]. Amplification of large mitochondrial DNA fragments before sequencing was carried out by Long-Range PCR, which minimizes the chance of amplifying mitochondrial pseudogenes from the nuclear genome (numts) [48,49]. We used a Long-Range PCR Kit (Qiagen, Inc., Valencia, CA, USA), with a reaction volume of 25 μL and a reaction mix consisting of 2.5 μL of 10× Long-Range PCR Buffer, 500 μM of each dNTP, 0.6 μM of each primer, 1 unit of Long-Range PCR Enzyme, and 50–250 ng of template DNA. PCR reactions were carried out in a Perkin Elmer Geneamp PCR System 9600 Thermocycler and a Bio-Rad ICycler. Cycling conditions were as follows: 94 °C for 5 min, followed by 45 cycles of denaturing at 94 °C for 30 s, then primer annealing at 50–57 °C (depending on primer set) for 30 s, and an extension at 72 °C for 8 min, followed by 30 cycles of denaturing at 93 °C for 30 s, annealing at 45–52 °C (depending on primer set) for 30 s, and extension at 72 °C for 5 min, with a final extension at 72 °C for 8 min. Four sets of primers [46] were used to generate overlapping amplicons from 3680 to 5011 bp in length, enabling a quality test for genome circularity [48]. Both mtDNA strands were sequenced directly using BigDye Terminator v3.1 (Applied Biosystems, Inc., Foster City, CA, USA). Sequencing products were analyzed on an ABI 3730 DNA Analyzer system (Applied Biosystems, Inc., Foster City, CA, USA). Sequences were assembled and edited using Sequencher 4.7 software (Gene Codes, Corp., Ann Arbor, MI, USA). Overlapping regions were examined for irregularities such as frameshift mutations and premature stop codons. A lack of such irregularities indicates an absence of contaminating numt sequences. We have not observed in any case any of these irregularities.

The alignments with all the mitochondrial amplicons were concatenated using Gblocks 0.91 [50,51] under a relaxed approach. The individual alignments were concatenated using the SequenceMatrix v1.7.6 software [52] to create a master alignment. Other software (Clustal X version 2.0; [53], and MUSCLE alignment plugin in Geneious R7.1; [54]) were employed to corroborate the alignments obtained and the results were always identical.

### 2.3. Nuclear DNA Microsatellites

Allele distribution was examined at six microsatellite loci (*Fca08*, *Fca43*, *Fca45*, *Fca96*, *Fca126*, and *Fca225*). Although microsatellites are not the best molecular markers for phylogenetic inferences, the microsatellite loci employed here showed a good phylogenetic signal in a previous study [55]. All loci consisted of dinucleotide repeats (CA)_n_ or (GT)_n_. The primer sequences for these microsatellites are reported elsewhere ([56,57]; see Appendix A). When DNA was extracted from skin or muscle, a polymerase chain reaction (PCR) was performed in a 25 μL volume. In this case, PCR reaction mixtures included 2.5 μL of 2.5 mM MgCl_2_, 2.5 μL of a 10× buffer, 1 μL of 1 mM dNTPs, 10 pmol of each primer, 14.5 μL of H_2_O, 2 μL of DNA (50–100 ng/μL), and one unit of Taq polymerase. DNA extracted from hair (10% Chelex resin; Walsh et al. 1991) was prepared for PCR as 50 μL reaction volume, containing twice the quantity of all the above reagents and 20 μL of DNA. PCR reactions were carried out in a Perkin Elmer Geneamp PCR System 9600 Thermocycler and a Bio-Rad ICycler. Cycling temperatures for all loci were as follows: 95 °C for 5 min, 35 cycles of 1 min at 95 °C, 2 min at 55 °C, 2 min at 72 °C, and a final extension of 5 min at 72 °C. Amplification products were kept at 4 °C until used. They were separated by electrophoresis in denaturing 6% polyacrylamide gels and then visualized in a Hoefer SQ3 sequencer vertical camera. Alleles were seized by comparison with a molecular size marker (ϕ174 digested with Hind III and Hinf I). A molecular weight marker was loaded every four lanes. We performed the PCR amplifications three times to ensure the accuracy of the genotypes obtained. In nearly 96% of the cases, the observed genotypes were the same in the three replicates. Thus, preferential allele amplification had minimal effects on our results.

### 2.4. Statistical Analyses

#### 2.4.1. Mitochondrial Genes

##### Phylogenetic Analyses and Temporal Split Estimations

jModeltest v2.0 [58], Kakusan4 [59], and MEGA X 10.0.5 software [60] were applied to determine the best evolutionary mutation model for the analyzed sequences for each gene, for different partitions, and for complete mitogenomes. All three programs offered identical results. The Akaike information criterion (AIC) [61,62] was used to determine the best evolutionary model for the relationships among the studied neotropical wild cats. The GTR + G model (General Time Reversible model + γ distributed rate variation among sites, [63,64]) was found to be the best nucleotide substitution model for the two data sets employed (full mitogenomes and the mt*ND5* gene).

Phylogenetic trees were constructed using Maximum Likelihood (ML) and Bayesian Inference (BI) methods for both data sets. The ML trees were constructed using the RAxML v.7.2.6 software [65]. Five hundred bootstrap replicates using GTR were estimated for topologic support [65].

BI trees were constructed in the BEAST v. 1.8.1 software [66]. Four independent iterations were run using three data partitions (codon positions 1, 2, and 3) with six Markov Chain Monte Carlo chains sampled every 10,000 generations for 40 million generations after a burn-in period of 4 million generations. We checked for convergence using Tracer v1.6 [67]. We plotted the likelihood versus generation and estimated the effective sample size (ESS > 200) of all parameters across the four independent analyses to determine convergence and optimal results. The results from different runs were combined using LogCombiner v1.8.0 and TreeAnnotator v1.8.0 software [68]. A Birth-death speciation model and a relaxed molecular clock with an uncorrelated log-normal rate of distribution were used [69]. Posterior probability values provide an assessment of the degree of support of each node on the tree. FigTree v. 1.4 software produced visualizations of the trees [70]. The BI trees were identical to the ML trees regarding the phylogenetic relationships of the Nariño cat with other *Leopardus* taxa. Additionally, the BEAST v1.8.1 program was run to estimate the time to the most recent common ancestor (TMRCA) for different nodes of the BI trees. We calibrated the dated tree by setting the diversification within *Leopardus* to 3.05 ± 1.1 million years ago (My) (97.5% confidence interval: 0.89–5.21 My). This value is the arithmetic mean of two previous temporal values for the diversification of the current *Leopardus* genus [29,33]. Although priors are ideally used in phylogenetic trees constructed with fossil information, the fossil record of the *Leopardus* lineage is fragmented and scarce. However, the fossils that have been studied are not in conflict with the molecular data we used [71].

The BI-dated tree belongs to one of two different approaches for inferring divergence times [72]. The first approach is based on fossil-calibrated DNA phylogenies. The second, the “borrowed molecular clocks” approach, uses direct nucleotide substitution rates inferred from other taxa. For this second approach, we used a median-joining network (MJN) with the help of Network 4.6.10 software from Fluxus Technology Ltd. [73]. The ρ statistic was estimated and transformed into years of divergence among the haplotypes studied [74]. To determine the temporal splits, we estimated the mutation rate per sequence and per million years. For the mt*ND5* gene, we used a mutation rate of 1.22% per My per position, which is equivalent to one mutation every 260,213 years [26,75]. For the entire mitogenome, we used an average mutation rate of 1.15% per My per position, which is equivalent to one mutation every 5500 years. This average mutation rate was obtained from several mt genes studied in felids (*ND5*, *ATP-8*, *ATP-6*, *16S rRNA*, *12S rRNA*, *Cyt-b*, and *COI*) [75,76]. The networks are more appropriate for intraspecific phylogenies than tree algorithms because they explicitly allow for the co-existence of ancestral and descendant haplotypes, whereas trees treat all sequences as terminal taxa [77].

##### Genetic Distances

We used the Kimura 2P genetic distance [78] to determine the percentage of genetic differences between the Nariño cat and all the *Leopardus* taxa analyzed, as well as with *F. catus* and *H. yagouaroundi*. We compared these genetic distances because values higher than 6–11% are typical for distinct species [79,80,81]. We employed this genetic distance because the Kimura 2P genetic distance is a standard measurement for barcoding tasks [82,83]. For this task, we employed the data for entire mitogenomes and for the mt*ND5* and mt*Cyt-b* genes.

#### 2.4.2. Nuclear DNA Microsatellites

##### Genetic Relationships among Neotropical Cat Species

For the microsatellite dataset, we conducted three analyses. (1) In the first one, we selected several specimens for each one of the species studied including the Nariño cat. This analysis involved the creation of a tree with Ward’s minimum variance method [84]. It was based on the squared Euclidean distance. In this analysis, we used 37 specimens (the Nariño cat, one *L. geoffroyi*, one *L. guigna*, four *L. jacobita*, five *L. tigrinus pardinoides*, five *L. pardalis*, five *L. wiedii*, four *L. c. cruscinus*, four *L. c. garleppi*, three *L. c. budini*, and four *L. c. steinbachi*). (2) Using the 113 felid specimens analyzed for microsatellites, we analyzed them by species (not as individuals as in the previous analysis). We constructed a tree with the unrooted unweighted neighbor-joining procedure (UUNJ) and with the Euclidean distance [85]. (3) Additionally, a second analysis by species we carried out was a Principal Coordinate Analysis (PCoA) using Gower’s procedure [86]. For this, we used a δμ^2^ genetic distance matrix among the species analyzed [87]. A minimum spanning tree (MST) was superimposed on the PCoA [88,89].

## 3. Results

### 3.1. Description of the Skin of the Nariño Cat

The skin of the Nariño cat (probably a female) has interesting characteristics. From a global perspective, this exemplar belongs to the tigrina morphotype I [45]. It has rosettes in oblique chains, yet these rosettes have fuzzy edges. The ground coloration is tawny-orange, but the dorsal crest is of a darker orange-brownish color. The tail is relatively short and is completely ringed, bearing seven complete rings and a black tip. However, this skin also has unique, diagnostic features. Its ground coloration is more reddish than in other *L. tigrinus* phenotypes. Most of the rosettes are bordered by black rims, but the rosettes’ interiors have a much more intense reddish color than that of other *L. tigrinus* specimens. Compared to other *L. tigrinus* exemplars the authors have examined, the top of the Nariño cat’s head and its dorsal crest are much darker. Its coat is denser and woollier. The head is rounder and wider, and the face is flatter. The body is short and relatively more robust than in other *L. tigrinus* taxa. Measurements from the skin were as follows; the total body length (including head) is 458 mm, head = 69 mm, length of left hind foot = 156 mm, width of left hind foot = 19 mm, length of left front foot = 109 mm, width of left front foot = 22, and tail = 280 mm.

The skin of the Nariño cat can be seen from different perspectives in Figure 1A–C. Figure 1D shows the Nariño cat (right) next to an *L. tigrinus* specimen from the Department of Caquetá (Colombia) (left). The difference in coloration and markings between the two is apparent. Figure 1E,F shows one individual sampled in Costa Rica, which has the same appearance as the holotype of *L. t. oncilla* (*F. p. oncilla*; type locality: Volcán de Irazu, Costa Rica), and one exemplar sampled in Intag (Imbabura Province, Ecuador). Both individuals were morphologically similar and had closely related mitochondrial haplotypes. Figure 1G includes a skin from Venezuela very similar to the holotype of *F. p. emerita* (type locality: Merida, Venezuela); Figure 1H,I shows the skin of the holotype of *M. caucensis* (type locality: Las Pavas, Cauca Department, Colombia), which is the tigrina-like morphotype found geographically closest to where the Nariño cat was discovered, and Figure 1J shows a skin very similar to the holotype of *M. t. elenae* (type locality: Santa Elena, Antioquia, Colombia). Nonetheless, none of these specimens closely resembles the Nariño cat.

Another holotype of tigrina is *F. p. andina* (type locality: Jima, Province of Azuay in southern Ecuador). We do have photographs of animals currently sampled in the same Ecuadorian province (Azuay Province) where *andina* was defined. Figure 1K,L shows the aspect of two exemplars from this area of Ecuador, and they are not similar to that of the Nariño cat.

We did not obtain photographs of the holotype of *F. carrikeri* (type locality: Pozo Azul, Costa Rica). However, *F. carrikeri* was synonymized with *F. p. oncilla*, and, therefore, their pelages should be similar to each other, and they are different from that of the Nariño cat.

Another *Leopardus* cat species (*L. c. thomasi*) inhabits the southern Colombian Andes close to the northern Ecuadorian border where the Nariño cat was collected. Its morphotype is also reddish and “a priori” similar to that of the Nariño cat. Figure 1M,N shows two exemplars from two different areas of Ecuador. However, the morphology of the Nariño cat is distinctly different from that of *L. colocola thomasi*.

The other small cats living in southern Colombia (*L. wiedii*, *L. pardalis*, *H. yagouaroundi*, and *F. catus*) have phenotypes that are distinctly different from that of the Nariño cat.

### 3.2. Mitochondrial Analysis

#### 3.2.1. Phylogenetic Analyses of Mitogenomes

The ML/BI trees constructed from the complete mitogenomes show the Nariño cat is the sister taxon of a clade comprised of Central American and trans-Andean tigrinas plus *L. guigna* and *L. geoffroyi* (ML tree: bootstrap 81%; BI tree: posterior probability = 1) (Figure 2).

The MJN procedure detected the same haplogroups as in the ML tree analysis. We detected one haplogroup with *L. guttulus* from southern Brazil and northeastern Argentina. The haplogroups of *L. guigna* and *L. geoffroyi*, were more related to the Central American *L. tigrinus* haplotype than to the trans-Andean Ecuadorian *L. tigrinus* haplotype. The two last haplotypes were separated by several undetected or extinct haplotypes. The Nariño cat was more closely related to, although differentiated from, some of these undetected or extinct haplotypes between the Central American and the trans-Andean *L. tigrinus*. Another haplogroup includes the ocelots and margays mixed with some northern Andean *L. tigrinus*. The remaining haplogroups were *L. jacobita*, *L. c. braccatus* (with hybridized tigrinas from northeastern Brazil), and the northern Andean Colombian and Ecuadorian tigrinas with no signals of hybridization with ocelots and margays, respectively.

Based on the dated BI tree of complete mitogenomes, the temporal split between the Nariño cat’s ancestor and the closest taxon (ancestor of Central American and trans-Andean *L. tigrinus* + [*L. guigna* + *L. geoffroyi*]) was 1.09 ± 0.02 My. The same split from the MJN procedure was 1.33 ± 0.01 My. The average of these estimates is around 1.2 My, coinciding with the beginning of the pre-Pastonian glacial period (1.3–0.8 My), and the Ensenadan South American mammalian fauna (1.2–0.4 My).

#### 3.2.2. Genetic Distance Analysis with Mitogenomes

The smallest Kimura 2P genetic distances between the Nariño cat and other *Leopardus* taxa were the Central American and trans-Andean tigrinas (5.6%) and *L. geoffroyi* (5.8%) (Table 1). These values were higher than other genetic distances found between well-recognized species, such as *L. guigna* and *L. geoffroyi* (2.0%), *L. pardalis* and *L. wiedii* (4.4%), *L. geoffroyi* and Central American and trans-Andean *L. tigrinus* (5.0%), or *L. guigna* and Central American and trans-Andean *L. tigrinus* (5.3%).

#### 3.2.3. Phylogenetic Analyses at the mt*ND5* Gene

The ML/BI trees with only the mt*ND5* gene data showed the Nariño cat is the sister taxon of *L. colocola*, including all the recognized subspecies of this species (87%/0.92). However, the mt*ND5* tree is less informative than the ML tree before and for brevity is not shown.

For the MJN procedure, we detected one haplogroup for some *L. tigrinus* individuals from Colombia and Ecuador. We also detected one haplogroup found in many Colombian and Venezuelan *L. tigrinus* mixed with margays and ocelots. Another haplogroup was that of *L. guttulus* from southern Brazil and northeastern Argentina. This last haplogroup also included central Brazilian individuals of tigrina, which were not mixed with *L. colocola*. Another haplogroup contained sub-haplogroups correlated with different morphological subspecies of *L. colocola* (*braccatus*, *colocola*, *wolffshoni*, *garleppi*, *pajeros, budini*) as occurred in the ML tree. This supports the point of view that all these subspecies comprise one unique species. Several northern Brazilian tigrinas (*L. t. tigrinus* or *L. emiliae* [45]) occurred within this *L. colocola* haplogroup because they showed introgression with the mtDNA of *L. c. braccatus*. The haplotype of the Nariño specimen was differentiated from those of other taxa. It was placed between the haplogroup of *L. colocola* and that of the Central American and trans-Andean *L. tigrinus*.

The dated BI tree, for the data for the mt*ND5* gene, showed that the temporal split between the ancestor of the Nariño cat from the most closely related taxon (the ancestor of *L. colocola*) was 1.94 ± 0.10 My. The MJN procedure detected the split to have occurred 1.82 ± 0.22 My. Thus, both temporal estimates were similar and suggest the split occurred around 1.9 Ma, coinciding with the beginning of the Calabrian age of the Pleistocene. This value was somewhat higher than the one found with the whole mitogenomic analysis.

#### 3.2.4. Genetic Distance Analysis at the mt*ND5* and mt*Cyt-b* Genes

At the mt*ND5* gene, the smallest Kimura 2P genetic distance between the Nariño cat and other *Leopardus* taxa occurred with *L. tigrinus*, which has mitochondrial DNA of *L. c. braccatus* (5.5%). Similar short genetic distances were observed between the Nariño cat and all taxa of *L. colocola* (5.6–6.3%) as well as with the Central American and trans-Andean *L. tigrinus* (6.7%). These values were higher, or similar, to other genetic distances found between well-recognized species, such as *L. guigna* and *L. geoffroyi* (2.4%) as well as *L. pardalis* and *L. wiedii* (5.0%). Other pairs with similar genetic distances include *L. geoffroyi* and the Central American and trans-Andean *L. tigrinus* (6.4%), or *L. guigna* and the Central American and trans-Andean tigrina (6.5%). The smallest genetic distances at the mt*ND5* gene were among all the taxa of *L. colocola* (1.2–2.9%). At the mt*Cyt-b* gene, the smallest Kimura 2P genetic distances between the Nariño cat occurred with the Central American and trans-Andean tigrinas (2.8%) and with *L. geoffroyi* (3.0%). Other small Kimura 2P genetic distances were with *L. colocola* (3.2%) and with the Colombian Andean tigrinas without hybridization with margays and ocelots (3.3%). Nonetheless, these values were higher than other genetic distances found between well-recognized species, such as *L. pardalis* and *L. wiedii* (0.3%), *L. wiedii* and *L. jacobita* (2.3%), *L. geoffroyi* and *L. guigna* (0.8%), the Central American and trans-Andean tigrina and *L. colocola* (2.0%), and the Colombian Andean tigrina without hybridization with margays and ocelots and *L. colocola* (2.1%). All the Kimura 2P genetic distances at the mt*Cyt-b* between the Nariño cat and all the other analyzed *Leopardus* taxa were higher than 2.5%, a value considered enough to differentiate between mammalian species [90]

### 3.3. Microsatellites

The Ward tree (Figure 3) showed that the Nariño cat was closely related to *L. geoffroyi* and *L. guigna*. In this tree, we detected three large clusters. One was composed of individuals of *L. pardalis*, *L. wiedii*, *L. t. pardinoides*, and *L. jacobita*. The second was composed of individuals of all the subspecies of *L. colocola*, and the third comprised *L. guigna*, *L. geoffroyi*, and the Nariño cat. The UUWNJ tree with Euclidean distances (Figure 4) also linked the Nariño cat with *L. geoffroyi* and *L. guigna* as well as the mitogenome analysis. This finding was also supported by the PCoA with the MST superimposed using the δμ^2^ genetic distance (Figure 5).

Therefore, all the molecular analyses, using both mitochondrial and nuclear markers, provided relatively similar results suggesting that the Nariño cat is a new and previously undetected cat taxon or species within the *Leopardus* genus.

## 4. Discussion

All the analyses carried out (mitogenomes, mt*ND5*, nuclear DNA microsatellites, as well as some pelage traits) clearly show the Nariño cat as a new taxon or species.

### 4.1. Some Taxonomic Insight of the Tigrina

Nascimento and Feijó [45] did not differentiate *L. t. tigrinus*, *L. t. pardinoides* and *L. t. oncilla* morphologically. These authors concluded that all the tigrinas from Central America, northern South America (Venezuela, Guianas, northern Brazil), northwestern and western South America (Colombia, Ecuador, Peru, Bolivia, and northwestern Argentina) belonged to one unique taxon, *L. t. tigrinus*; this correlated with their morphotype I. In contrast, one molecular genetic work [46] and this paper indicate that there are at least four distinctly different tigrina mitochondrial lineages occurring in Central America, northwestern and southwestern South America. One distinct lineage is the Central American and trans-Andean Colombian and Ecuadorian tigrina. The second lineage is present in the Andean tigrina with the mtDNA of this lineage differentiated from that of the margays and ocelots. The third lineage is also within the Andean tigrina, but this lineage has mtDNA very similar to those of margays and ocelots. This similarity may be the product of hybridization between these species or old introgression of the ancestors of margays and ocelots within the tigrina. The fourth lineage is the Nariño cat. Therefore, morphotype I [45] is composed of four significantly different molecular lineages, and these different lineages could represent three or four different species, at least, if we employed the PSC [1,91]. Identically, Kitchener et al. [92] concluded (excluding *L. guttulus* from *L. tigrinus*), taking into consideration the results of Li et al. [33], the existence of two subspecies of *L. tigrinus* in the Neotropics: *L. t. oncilla* in Central America, and *L. t. tigrinus* in South America. Our results show that these studies do not adequately represent the real number of taxa or species within the tigrina complex. Contrarily, old works [93] agree quite well with our new molecular results claiming the existence of at least three or four different species of tigrina in Central America and in northwestern South America.

Furthermore, the recent morphological work detected highly similar pelages among some tigrinas and the margay [45]. There were four cases of pelage similarity between these two species from Venezuela, Guyana, El Tambo, in the Cauca Department (Colombia), and Las Pavas, Valle del Cauca Department (Colombia). This coat similarity could be related to hybridization between some tigrinas and margays that have been previously detected [46] and in this study.

Another relevant finding of this paper is that all the analyzed specimens of *L. colocola* represent a single species although their phenotypes were heterogeneous. Thus, our results do not support *L. colocola* comprising three differentiated species (*L. colocola*, *Leopardus pajeros*, *L. braccatus*) [34].

However, the most striking discovery of the present study may be that the Nariño cat has a clear molecular differentiation from all other known taxa in the genus *Leopardus* as well as some pelage characteristics that distinguish it from all tigrina taxa previously described; therefore, the Nariño cat needs to be described as a new species or taxon within the *Leopardus* genus.

### 4.2. The Nariño Cat

From a morphological perspective, none of the holotypes of the defined forms of tigrina throughout Central America and northwestern South America (*F. pardinoides*, *F. p. oncilla*, *F. carrikeri*, *F. p. emerita*, *F. p. andina*, *M. t. elenae*, and *M. caucensis*) closely resemble the Nariño cat. Some of the characteristics of the Nariño cat’s skin, e.g., rosettes (with edges clearly black but intensely reddish inside), are similar to those we observed in individuals of the *andina* form (Azuay, Ecuador). However, for all other pelage characteristics, the Nariño cat was differentiated from *andina*. The other small spotted cats in the southern Colombian Andes and in the northern Ecuadorian Andes are the margay, the ocelot, and the colocolo cat. The pelages of margays and ocelots are differentiated from that of the Nariño cat. The comparison between the colocolo and the Nariño cat is interesting because there are similarities in some morphological characters. Both taxa have more rounded and squatter bodies than the tigrinas. In addition, their coats are redder, and their heads are broader and rounder in comparison to those of the tigrinas. However, some characteristic traits of the colocolo do not occur in the Nariño cat. For example, the colocolo cat has completely red rosettes, including their edges. There is also a black-brownish transverse stripe on the upper part of the chest, and the colocolo cat also has two to four contrasting black stripes on the upper part of the front legs. These characteristics are absent in the Nariño cat. Indeed, one specialist in morphology of *L. colocola* (Dr Rosa García-Perea, Museo Nacional de Ciencias Naturales, Madrid, Spain) commented, “…This rare individual is not a *L. colocola*. It should be one of these strange tigrinas, which occasionally appear…”

The mitogenomic analysis showed the Nariño cat is the sister taxon of the clade formed by the Central American and trans-Andean tigrinas + *L. geoffroyi*/*L. guigna*. Based on the nuclear DNA microsatellite analysis, the Nariño cat is the sister taxon to the clade composed of *L. geoffroyi* + *L. guigna* (the Central American and trans-Andean tigrinas were not analyzed for microsatellites). Therefore, both the mitogenomic data and the nuclear microsatellite data offered identical results regarding the relationships of the Nariño cat with two small spotted cats (*L. geoffroyi* and *L. guigna*). These two small spotted cats are morphologically dissimilar to and distributed more than 3500 km away from the southern Colombian Andes where the Nariño cat’s type locality. Therefore, the Nariño cat could be a subspecies of *L. oncilla* (this last taxon elevated to the category of species), or it is a new sister species to the Central American and trans-Andean tigrinas (*L. oncilla* + (*L. geoffroyi*/*L. guigna*)). Additionally, the Nariño cat is not some form of hybrid of known cat species (of the tigrina lineages, margay, ocelot, colocolo, jaguarundi, or domestic cat) from that area of Colombia. Its mtDNA differs from that of every known cat species. If the Nariño cat was a hybrid, the female lineage of this Nariño cat specimen is undoubtedly a different species, which is unrecognized until now by the scientific community.

Because the Nariño cat has a different coat pattern from those of other tigrina forms and other *Leopardus* species, and because the Nariño cat is genetically (nuclear and mitochondrial genes) distinct from other *Leopardus* taxa, we suggest the Nariño cat as a new species:

*Leopardus narinensis* sp. nov. Ruiz-García 2018. URN: LSID: ZOOBANK.ORG: ACT: 037B9EF0-8AEF-4717-8C5A-56D9E293C6DF.

Etymology: The specific name refers to the Nariño Department in southern Colombia where this specimen was obtained. We propose the common name of Nariño or Galeras cat (by its origin) or red tigrina (because its pelage is mostly reddish). If the geographical distribution of this new taxon is larger than is currently believed, red tigrina would be the preferred common name.

Holotype: The holotype skin is in the Instituto von Humboldt (ID 5857) (Figure 1).

Type locality: The specimen was collected in 1989 on the Galeras Volcano, Nariño Department (Colombia) (1°13′43.8” N; −77°21′33” W), 3100 m above sea level (masl).

Morphological diagnosis: Its ground color is more reddish than in other *L. tigrinus* phenotypes. Most of the rosettes are bordered by black rims, but the rosettes’ interiors have a much more intense reddish color than that of other *L. tigrinus* specimens. Compared to other *L. tigrinus* taxa, the top of the Nariño cat’s head and its dorsal crest are much darker. Its coat is denser and woollier. The head is rounder and wider. The body is relatively more robust than in other *L. tigrinus* taxa.

Molecular diagnosis: Species-level diagnostic characters for the Nariño cat were observed in the two mitochondrial genes for which there are more sequences in neotropical cats (mt*ATP8* and mt*ND5*) and yielded the following seven synapomorphies: At the *ATP8* gene (three synapomorphies) in the nucleotide positions 8530 (T), 8594 (A), and 8597 (T). At the *ND5* gene (four synapomorphies) in the nucleotide positions 12,506 (A), 12,715 (A), 12,737 (A), and 12,749 (A). These seven synapomorphies differentiated these two mt genes of the Nariño cat from those of all the other species of the genus *Leopardus* and from the domestic cat and the jaguarundi.

Another important question is when the branch of the Nariño cat diverged from other *Leopardus* lineages. Our temporal divergence splits agree with estimates carried out by other authors for *Leopardus*. Our initial estimate of temporal diversification in *Leopardus* (3.91 to 3.02 Ma) was very similar to that estimated by Li et al. [33], 3.97–3.14 My, and by Johnson et al. [29], 2.91 My. Therefore, the initial diversification in the genus *Leopardus* occurred during the Pliocene (Zanclean-Piacenzian ages). The cold and dry climate during the Pliocene coincided with the onset of high-latitude glacial cycles. Additionally, this Pliocene period agrees quite well with the last uplifting phase of the Andes [94] (see, for instance, the rising of the “tablazos” of Piura, Peru) and elevated volcanic activity in the Andes that led to a replacement of rainforests with steppe and grassland environments.

The temporal split of the Nariño cat’s ancestor from the ancestors of other *Leopardus* taxa occurred ~1.3–1.0 Ma, according to the analysis of the entire mitogenome. This possible scenario of differentiation of the ancestor of the Nariño cat could have been initiated by the pre-Pastonian glacial period (1.3–0.8 My; within the Calabrian age), which had the highest glacial peak of the first Quaternary glacial period (Günz). This glacial period was extremely dry, and there was a great degree of forest fragmentation. The Andean forests were transformed into open cold dry savanna (‘paramo’), which could have potentially isolated populations of different species. The Nariño cat was sampled in a “paramo” area. It was demonstrated that the mean temperature in the Colombian Andes was 4 °C lower than today [95]. At 2500 masl, the temperature was 10 °C lower than it is currently, and, further, precipitation was less than the level reported for today [95]. The area where the Nariño cat was sampled corresponds to the North-Andes Biogeographic province, in an endemism highland zone named “Nudo de los Pastos” characterized by a cold, or very cold, climate with a volcanic geomorphology origin [96]. This was also a time of differentiation for many carnivores, as it was previously determined for the colocolo cat and the foxes of the genus *Lycalopex* [97,98]. Around 1.3 My, the fauna of Buenos Aires transformed into a typical semi-arid Patagonian fauna, represented by the guanaco, *Lestodelphys* and *Lyncodon*. Therefore, the climate was considerably colder and drier than today and could have influenced the speciation process of the Nariño cat.

This new taxon is absent in the Latin American museums that we revised (in Colombia, Ecuador, Peru, Bolivia, Chile, Argentina, Paraguay, and Uruguay). In the wild, this taxon has not been recorded. Camera traps (since 2018 until now) in southern Colombia and northern Ecuador have yet to record the animal. This new taxon may be near-extinct or totally extinct. Henceforth, the specimen that we analyzed should be one of the last living exemplars of this taxon.

The following steps are needed to validate the existence of this new small spotted cat in the Neotropics. First, confirm that the taxon is not extinct and that there are still living individuals. Second, locate additional skins and skulls similar to the Nariño cat in collections or museums. The publication of this new taxon may create awareness to help discover new specimens to test the hypothesis raised here. Third, Illumina dye generation sequencing, 454 Life Sciences pyrosequencing, Pac-Bio, or Oxford Nanopore may determine the characteristics of the nuclear genome of both the Nariño cat and of all the tigrina taxa. These analyses would help us to determine the degree of hybridization of these small spotted cats with each other, as well as the degree of hybridization with other morphologically differentiated and well-recognized species of the *Leopardus* genus.

## Figures and Tables

**Figure 1 genes-14-01266-f001:**
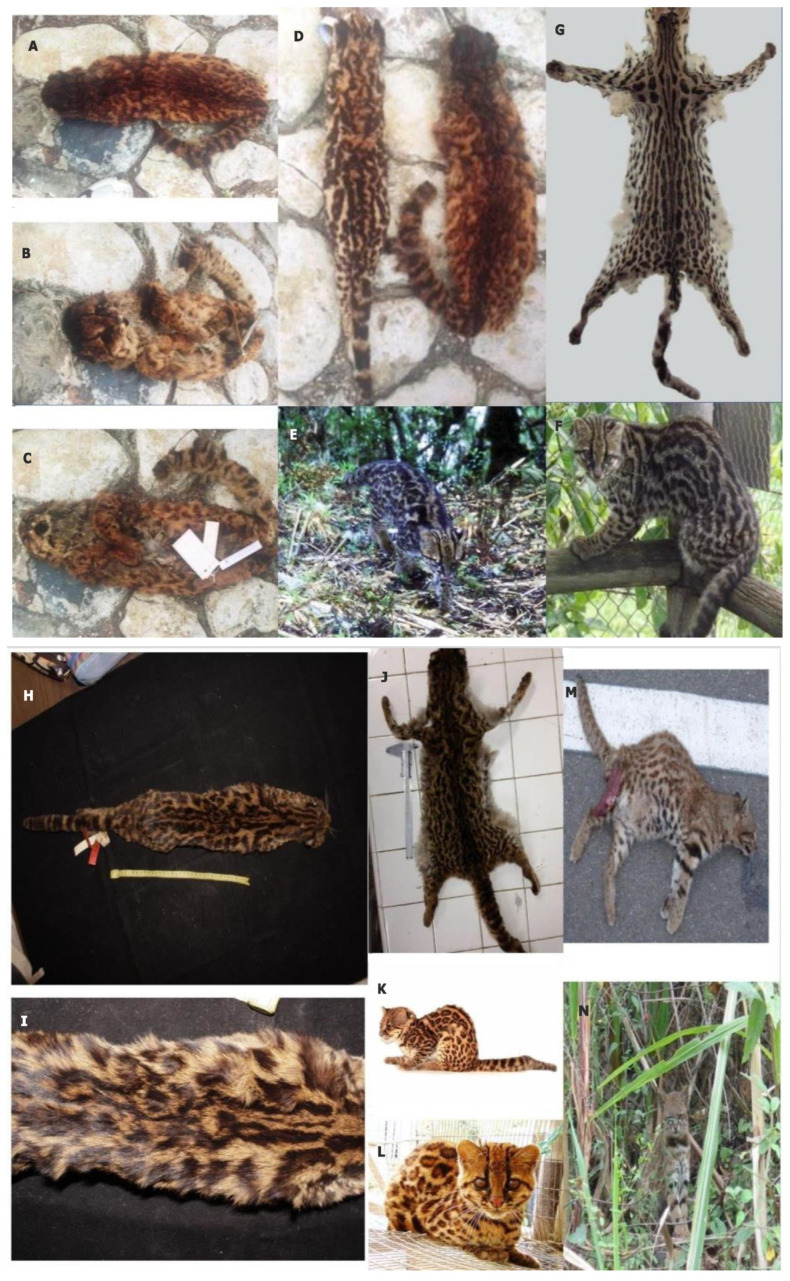
The Nariño cat found at the Galeras Volcano in the Nariño Department from the southern Colombian Andes and aspect of other specimens of other tigrinas from different regions of the Neotropics. (**A**–**C**) Different views of the new species *Leopardus narinensis*, Ruiz-García, 2018. Its morphology and its mitochondrial and nuclear microsatellite DNA do not coincide with that of any known species of the *Leopardus* genus in Latin America. (Photos Manuel Ruiz-García). (**D**) Comparison of the Nariño cat (right) with an individual of tigrina (from the Caquetá Department in Colombia) with a larger range in the northern Andes and with similar mitochondrial haplotypes to those of margays and ocelots (left) (Photo Manuel Ruiz-García). Central American and trans-Andean tigrinas: (**E**) One of the tigrinas sampled in Costa Rica (Photo José González-Maya), and (**F**) A tigrina analyzed from Intag (Imbabura Province, Ecuador) that had a phenotype very similar to the Costa Rican tigrinas that were analyzed and shown in the previous photo. Both tigrinas were molecularly confirmed as well-defined tigrina taxa or lineages. (Photo Manuel Ruiz-García). (**G**) Photo of a skin of a tigrina from Venezuela that was very similar to the holotype of *Felis pardinoides emerita* Thomas, 1914 (type locality: Merida, Venezuela). (Photo Anderson Feijó). (**H**,**I**) Photos showing the holotype of *M. caucensis* Allen, 1915 (type locality: Las Pavas, Cauca Department, Colombia). (Photo Anderson Feijó). (**J**) Photo of a skin that was very similar to the holotype of *M. tigrina elenae* Allen, 1915 (type locality: Santa Elena, Antioquia, Colombia). (Photo Manuel Ruiz-García). (**K**,**L**) Photos of two tigrinas sampled in the Azuay Province (Ecuador) where the holotype of *F. p. andina* Thomas, 1903 (type locality: Jima, Province of Azuay in southern Ecuador) was originally discovered. (Photo Juan Carlos Sánchez and Photo Jorge Brito, respectively). Photos of *L. colocola* from Ecuador with a differentiated morphology (plus differentiated mitochondrial DNA and nuclear microsatellite alleles) from a geographical area near where the Nariño cat was discovered: (**M**) Specimen of *L. colocola* killed in San Lorenzo in the Imbabura Province, northern Ecuador (Photo Diego Tirira), and (**N**) Exemplar of *L. colocola* from Macará, southern Ecuador (Photo Diego Tirira). The morphologies of these two specimens (**M**,**N**) differ from that of the proposed new species, *L. narinensis*.

**Figure 2 genes-14-01266-f002:**
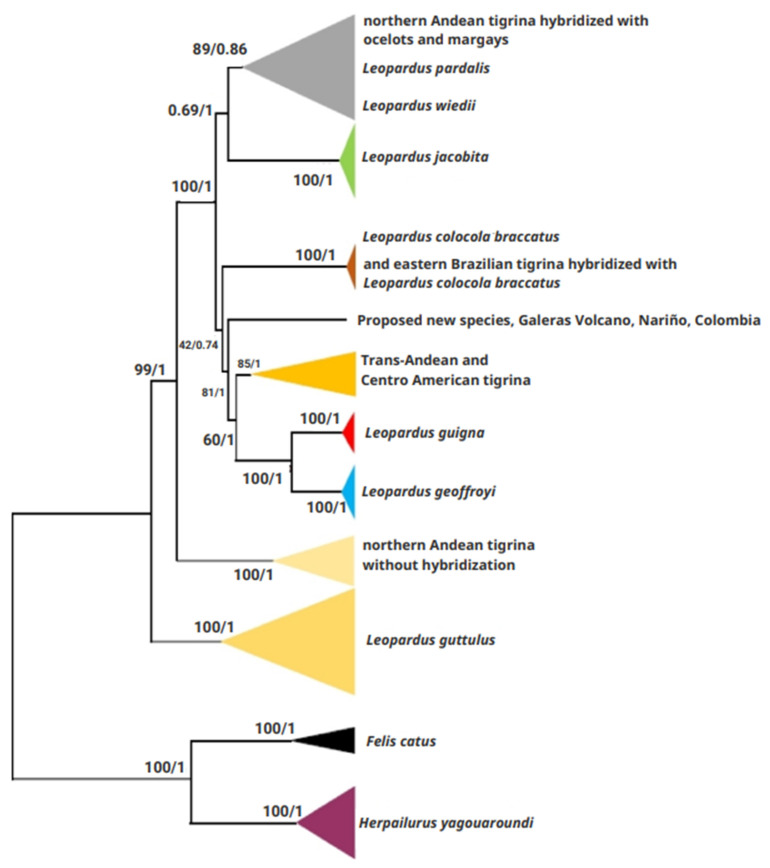
Maximum likelihood and Bayesian inference trees of 44 felid specimens, including 18 tigrina-like morphotype specimens analyzed for their entire mitogenomes. Six different clades include tigrina-like specimens: *L. emiliae* with mtDNA of *Leopardus colocola braccatus*; the Nariño cat; *L. guttulus*; the Central American and trans-Andean tigrina; the Andean tigrina; and the Andean tigrina with mtDNA closely related to that of margays and ocelots. The sequences of other species of *Leopardus* were included. Sequences of *F. catus* and *H. yagouaroundi*, two species were also included which also live in the same geographical area where the Nariño cat was sampled. The first number at the nodes indicates bootstrap support (%) (ML tree); the second number at the nodes indicates posterior probability (BI tree).

**Figure 3 genes-14-01266-f003:**
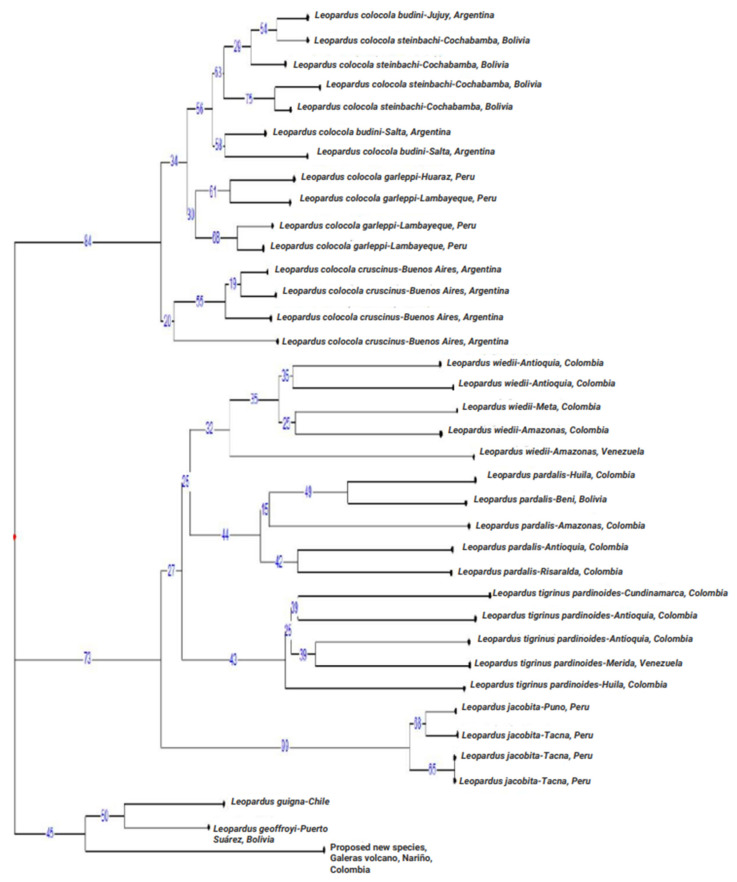
Ward tree of squared Euclidean distances among 37 specimens, selected from the 113 specimens studied of the genus *Leopardus*, using data from six nuclear DNA microsatellites. These 37 specimens were the Nariño cat, 15 *L. colocola* (two *budini*, three *steinbachi*, two *budini*, four *garleppi*, and four *cruscinus*), five *L. wiedii*, five *L. pardalis*, five *L. t. pardinoides*, four *L. jacobita*, one *L. guigna*, and one *L. geoffroyi*. Numbers at the nodes indicate bootstrap support (%).

**Figure 4 genes-14-01266-f004:**
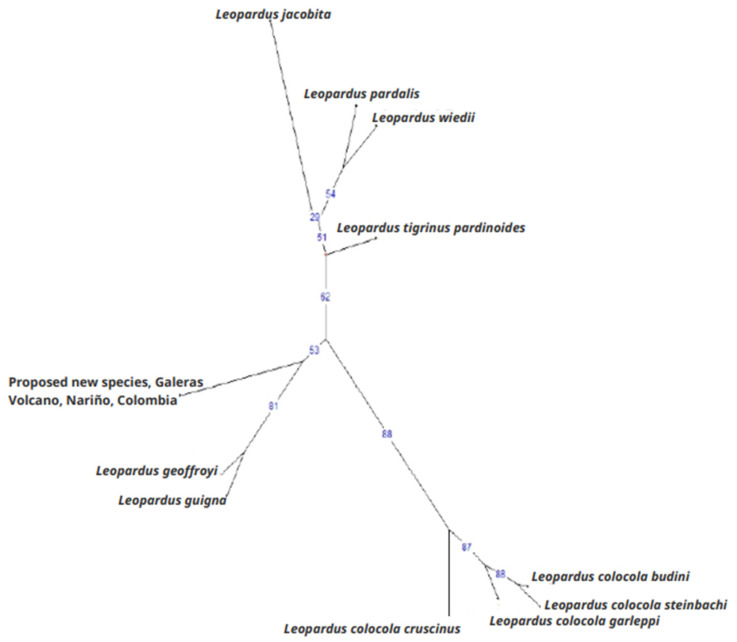
Tree from the unrooted unweighted neighbor-joining procedure (UUNJ) with Euclidean distances from six nuclear DNA microsatellites. The species of *Leopardus* analyzed were the Nariño cat, *L. tigrinus pardinoides*, *L. wiedii*, *L. pardalis*, *L. jacobita*, *L. guigna*, *L. geoffroyi*, and *L. colocola* (*cruscinus*, *budini*, *steinbachi*, and *garleppi*). Numbers at the nodes indicate bootstrap support (%).

**Figure 5 genes-14-01266-f005:**
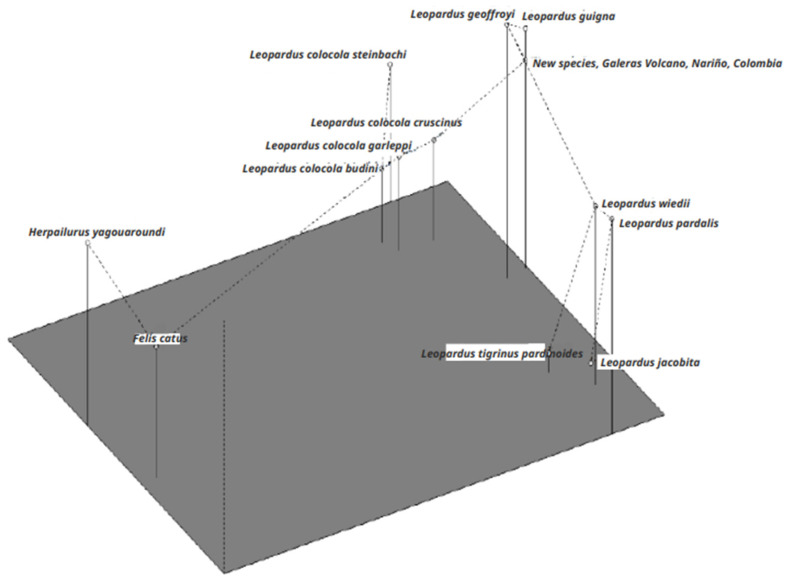
Principal Coordinate Analysis (PCoA), with Gower’s procedure, and with the δμ^2^ genetic distance matrix among eight species of the *Leopardus* genus, including the Nariño cat, *F. catus*, and *H. yagouaroundi*. A minimum spanning tree (MST) was superimposed on the PCoA. The species of *Leopardus* analyzed were the Nariño cat, *L. t. pardinoides*, *L. wiedii*, *L. pardalis*, *L. jacobita*, *L. guigna*, *L. geoffroyi*, and *L. colocola* (*cruscinus*, *budini*, *steinbachi*, and *garleppi*).

**Table 1 genes-14-01266-t001:** Kimura 2P genetic distances for entire mitogenomes among the different species of the *Leopardus* genus, including the Nariño cat, and different subspecies of *L. colocola*. Below the main diagonal, genetic distance values are in percentages (%). Above the main diagonal, standard errors in percentages. 1 = Andean tigrina, *L. tigrinus*, introgressed by margays and ocelots; 2 = Andean tigrina*; L. tigrinus*, not introgressed by margays and ocelots; 3 = *L. guttulus*; 4 = Central American and trans-Andean tigrina; 5 = *L. emiliae*; 6 = *L. c. braccatus*; 7 = *L. guigna*; 8 = *L. wiedii*; 9 = *L. pardalis*; 10 = *L. geoffroyi*; 11 = *L. jacobita*; 12 = Nariño cat (*L. narinensis*); 13 = *F. catus*; 14 = *H. yagouaroundi*.

Cat taxa	1	2	3	4	5	6	7	8	9	10	11	12	13	14
1	-	0.9	1.2	0.8	0.9	0.9	0.8	0.4	0.3	0.8	0.7	0.7	1.5	1.6
2	10.6	-	1.4	1.2	1.3	1.3	1.3	1.0	1.0	1.3	1.2	1.1	1.8	2.0
3	15.0	16.9	-	1.1	1.3	1.3	1.2	1.2	1.2	1.2	1.2	1.2	2.1	2.1
4	7.3	13.2	13.3	-	0.9	0.9	0.7	0.7	0.9	0.7	0.8	0.7	1.5	1.7
5	6.4	13.6	16.3	6.9	-	0.0	1.0	1.0	1.0	0.9	1.0	0.9	1.7	1.9
6	6.4	13.6	16.3	6.9	0.0	-	1.0	1.0	1.0	0.9	1.0	0.9	1.7	1.9
7	6.8	14.7	15.8	5.3	7.2	7.2	-	0.8	0.9	0.5	0.9	0.9	1.7	1.9
8	3.3	11.9	15.2	7.2	7.7	7.7	7.2	-	0.5	0.8	0.8	0.7	1.6	1.6
9	2.8	10.6	15.4	8.0	6.9	6.9	7.5	4.4	-	0.9	0.8	0.7	1.5	1.7
10	6.8	14.3	15.0	5.0	6.5	6.5	2.0	7.2	7.3	-	0.8	0.8	1.5	1.8
11	5.7	12.8	15.3	7.7	6.9	6.9	6.9	6.4	6.3	6.7	-	0.9	1.6	1.9
12	6.7	13.4	14.4	5.6	6.1	6.1	6.1	7.0	7.6	5.8	7.0	-	1.6	1.7
13	20.8	27.1	29.2	22.3	21.7	21.7	23.1	22.4	21.7	22.4	21.6	21.1	-	1.1
14	25.6	32.8	32.4	24.9	25.9	25.9	26.8	25.2	26.6	26.1	26.3	25.1	10.9	-

## Data Availability

The data sets generated and analyzed during the current study are available from the corresponding author on reasonable request at the e-mails, mruizgar@yahoo.es, and mruiz@javeriana.edu.co, and in the Appendix A, which are available on the online version of this paper. The GenBank accession numbers of the Nariño cat and the other *Leopardus* specimens herein analyzed are from MG230196.1 to MG230251.1.

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
