# Peer review of "Morphological and Genetics Support for a Hitherto Undescribed Spotted Cat Species (Genus Leopardus; Felidae, Carnivora) from the Southern Colombian Andes"

_genes, 2023, doi:10.3390/genes14061266_

Round 1
Author Response
Dear Editor of Genes,
Herein, we enclose all the corrections, changes carried out, and answers to the referee’ comments. We thank to the three referees for their respective opinions, especially to the referees 1 and 3 because all their observations helped to enhance the quality of the manuscript.
Referee 1:
Referee 1 said:
The authors describe a new small cat species from South America, specifically a new spotted cat species from the genus Leopardus. They support their claim by three lines of arguments, which comprise morphological (fur) features (based on a recently discovered skin in a museum), as well as mitochondrial and nuclear DNA divergence analyses. To exclude the possibility that the recently discovered skin simply belonged to a hybrid between sympatrically occurring cat species (e.g. between Leopardus cat species themselves and between Leopardus cats and feral domestic cats and/or P. yagouaroundi), the authors included all species that were potential candidates for hybridizations in their analyses as well and could show that this specimen is not a hybrid. The methodological procedures are solid, so that the reader can assess if the conclusions are truly warranted. The paper is essentially understandably written, but should be polished.
We thank this comment, and, effectively, we polished the manuscript extensively following his/her comments. All the blue corrections in the manuscript are those following to this referee, meanwhile all the red corrections belong to Referee 3.
Referee 1 said:
I also have some reservations toward the interpretations of the ND5 gene analysis. I understand why it was done (because not for all species that needed to be included, mitogenomes were available), however, this is only a small locus (which was selected opportunistically and not on scientific grounds because it had the highest number of species covered by genebank entries) and results from this analysis are only to some extend indicative but should not be overemphasized (rather toned down). The main message comes from Figures 6 (see my comments to Fig. 6 below) and 9, they need to be in the main manuscript. Figures 7 and 8 can be omitted or should be in the supplements.
We have not totally eliminated the results obtained exclusively with the mtND5 gen, because it is the unique gen in GenBank for which all the Leopardus taxa have sequences and therefore we compare all them with the Nariño cat. However, we have reduced many comments about this gene. For example, we deleted the Table 2, which contained the Kimura 2P genetic distances among all the species pairs with mtND5. In the Discussion, we also deleted the temporal hypothesis related with mtND5. On the other hand, we maintained Figures 7 and 8 (now Figures 3 and 4) by two reasons. First, it is important to show to the readers the unique analysis where all the Leopardus taxa are enclosed (mtND5), especially all the taxa within L. colocola due to that “a priori” we believed that the Nariño cat could be related to L. colocola. Second, Figure 8 (now Figure 4) shows the unique analysis that we carried out with individual specimens for the species analyzed with nuclear microsatellites. It is important to compare the microsatellite results for specimens and for species (Figures 9 and 10; now Figures 5 and 6) to show that in whatever case, the Nariño cat is not specially related with known tigrinas independently of specimens, species, number of taxa, markers, and mathematical procedures employed.
Referee 1 said:
I could not assess the supplementary figures S4 and 5 because the resolution quality was too low for these figures. However, for the assignment of species level to a new specimen, haplotype networks are not useful anyway as they carry (intrinsically) no phylogenetic information (they don’t have a root, they don’t have substitution models). I recommend removing this completely.
Following to this referee, we have deleted Figures S4 and 5. Some comments on the haplotype networks have also been deleted, but we have conserved one thing of the MJN procedure (haplotype networks), the temporal split among the ancestor of the Nariño cat and all the other Leopardus species to compare it with the temporal splits obtained by using Bayesian procedures. The temporal split results were very similar with these two well-differentiated methods, which is a solid argument to determine what climatic or geological events could produce the speciation event for the Nariño cat.
Referee 1 said:
All in all, the paper is way too long for the overall goal of species assignment to a newly discovered skin. A concise comparison of morphological and genetic dissimilarities with closely related species (to warrant species assignment) is strongly recommend. There is a reference list of 106 publications! Considering the reductions recommended above and in the comments below, this reference list can also be shortened significantly.
We have tried to shorten the manuscript (around 2 pages), but it is not easy without to lose important information. Furthermore, the referee should think that this manuscript is not only about …of species assignment to a newly discovered skin… It is also about the systematics of the tigrinas. Thus, we have reduced something the manuscript but not drastically.
Referee 1 said:
Specific comments:
Title (1): please consider rephrasing the title. The insert “nuclear and mitogenomics” is unusual (because the two terms act at different levels) and should not be used. If the authors want to keep it (although I think it is not needed), then it should be something like “nuclear and mitochondrial markers”. Corrected
Title (2): please avoid redundancy and consider revising the phrase “new undescribed”. (e.g.: hitherto undescribed, newly discovered, hitherto unknown, …) Corrected
L34: Please exchange “due” by “because” Corrected
L64: please change to “It integrated four subspecies” Corrected
L68: please remove “such” Corrected
L69: please remove “as” Corrected
L51-94: very detailed description of the taxonomic status. I suggest supplementing this section with a concise table summarizing this information. It would help the reader to keep the “big picture” in mind. Corrected
L97: please insert a noun after “chemical” (e.g.; compound, repellent, preservative, …) Corrected
L98: please change to “from this skin” Corrected
L99: please replace “obtained” by “available” Corrected
L100: please rephrase “traditional aspects”. I am not quite sure what the authors mean, perhaps “morphological traits”? Corrected
L103: please rephrase. The statement should be the other way around, namely that 112 specimens were genotyped at six microsatellite loci Corrected
L105: Sentence should end after “… were not analysed.” Corrected
L110: please change to “From the unknown felid species only a skin specimen was available (Instituto ….). Corrected
L120: please use past tense and change “contains” to “contained” (same for L122) Corrected
L120: “collecting mitochondrial genomes” sounds weird. Please rephrase, e.g. “We sequenced complete mitochondrial genomes from…” “Additionally, we also retrieved full mitogenome sequences from Genbank (…).” Corrected
L123: Please replace “major” by “highest”. (One could also say: the majority) Corrected
L125-126: Please rephrase the following sentence “The GenBank accession numbers of the Nariño cat and the other Leopardus specimens herein analyzed are from MG230196.1 to MG230251.1”. E.g.: “All newly sequenced mitogenomes including the one from the Nariño cat were submitted to Genbank (acc nos.: MG….).” Corrected
L127 (in table 1): please explain what is meant by “undetermined”. Do you mean unknown origin? Corrected
L133: please change to: Additionally, we genotyped 133 cat specimens (comprising XY species) at six microsatellite loci … Corrected
L133-148: there is some redundancy here. This paragraph is followed by two methodological paragraphs regarding mtDNA and microsatellite loci. The information from lines 133-148 should be included in these two paragraphs. Corrected
L139-141: here (all of a sudden) appear new forms that were not mentioned in the introductory taxonomic survey. Corrected
L145: please insert an “also” (“We also analysed …”). In addition, as F. catus and P. yagouaroundi are mentioned here for the first time, their names should be written in full (Felis catus, Puma yagouaroundi). Corrected
L147: Please rephrase “… hybridize with the Leopardus species (L. t. pardinoides, L. wiedii, L. pardalis, and 147 L. colocola) that inhabit this geographical area.” to: “… hybridize with Leopardus t. pardinoides, L. wiedii, L. pardalis and L.colocola.” Corrected
L150: Please remove the half-sentence: “For the mitogenomic (including the fragment of mtND5 gene, 315 base pairs, bp, in 150 length) analysis (a total of 16,756 bp),… “. The information regarding the alignment length belongs to the results. The sentence should start with: “We extracted DNA from …”. Corrected
L152: please rephrase to: “DNA extraction from muscle fibers followed…” Corrected
L153: redundancy. Please remove “following the manufacturer’s protocol.” Corrected
L154: Please rephrase to: “For some skin samples optimized DNA extraction procedures had to be employed (ref.).” Corrected
L155: “Mitochondrial genomes were sequenced…“. This statement is not correct. Should be rephrased to: “Amplification of large mitochondrial DNA-fragments prior to sequencing was carried out by Long Range PCR … (refs.). We used a Long Range PCR kit (Qiagen Inc.) …” Corrected
L175: unclear what is meant by “alignment of all genes”. Alignment of mitogenomes? Please be more precise. Corrected
L181: the sentence “Microsatellites were examined” sounds odd. Please change to something along the lines of “Allele distribution was examined at six microsatellite loci.” I also suggest including the reference for the loci here (it comes later but it should rather be here) Corrected
L183: please change to “… the microsatellite loci employed here…” L184: Please consider changing to: “All loci consisted of dinucleotide repeats” Corrected
L187 (please check the entire paragraph): please either remove “PCR” or “reaction” in front of “volume” (should either be “reaction volume” or ”PCR volume”, but not “PCR reaction volume” as the “R” in PCR already stands for “reaction”) Corrected
L193: please change to “for all loci” to avoid redundancy (the section title already says that this section is all about microsatellite loci) Corrected
L195: I am not sure if the word “electrophoresed” is a valid word. I recommend using “electrophoretically separated” (or “separated by electrophoresis”). I would also remove the four words “used. PCR products were” from that sentence and would replace it by “they were electrophoretically separated …” Corrected
L197: the sentence sounds odd. Perhaps: “Alleles were seized by comparison with a molecular size marker (phi174, digested with HindIII and Hinf I). Question: did you prepare the size marker yourself or was it bought? Then please name the company. Corrected
L202: there are no “procedures” included in supplementary table 1. But please consider rephrasing anyway because sentences like this should be avoided. There should be a scientific statement and then in brackets where additional explanatory information can be found like “….. (suppl Table S1).” Corrected
L212-215: unclear on what sequences the ML analysis was performed. Was this performed on the full mitogenome alignment or just on ND5 gene? Please specify. [the authors give this information pages later with figs. 6 and 7, but it should be mentioned here already] Corrected
L214-215: please correct to “codon positions” instead of “just codon” L215: the previous paragraph (L206-210) does not reveal what model was actually the best fitting one. RaxML only provides a very limited model selection, but the reader does not know if the GTR+G model applied in RaxML was actually the closest one to the model found before. Corrected
L219: again, not clear what data sets were used, please specify. For the full mitogenome, not every sequence (coding and non-coding) is a protein coding gene. Were only protein-coding sequences from the mitogenome used? Corrected
L220: again, please correct to “codon positions” instead of “just codon” Corrected
L224: really of “all parameters?” Then it is quite unlikely that 40 million generations were enough (in particular for the posterior for the substitution rates) Corrected
L230: I don’t understand “for brevity”. If both tree reconstructions (ML and BI) resulted in the same tree topology, then the trees (Figs. 6 and 7) may as well show both values, the ML bootstrap support and the BI posterior. Corrected
L232: this sounds somewhat odd even though the explanation is then given on Line 234. I suggest to use a phrase like “We calibrated the dated tree by setting the split between X an Y to Z years ago.” Corrected
L239: were unusual phrasing (“BI temporal estimates”), normally this is called a dated tree. Corrected
L244: A median joining network is not needed to do so But a MJN is needed to compare the temporal split between the ancestor of the Nariño cat and other Leopardus ancestors with the estimates obtained with the Bayesian tree and thus some the MJN has some utility in this work
L246: All of a sudden, we are dealing with the ND5 gene only. This has not been mentioned before Corrected
L247: this is per million years per position Corrected
L258: this a very week argument because within species distances of ~16% have been reported (Sable antelope Hippotragus niger) and no systematic reviews on thresholds for species delineating genetic distances have been carried out yet. In this aspect, we disagree with the Referee 1. There are several works relating the values of genetic distances and the differentiation of species and other taxa levels. For example, Kartavtsev (2011) analyzed sequences of mt COI from 20,731 vertebrate and invertebrate animal species and obtained 0.89% ± 0.16% for populations within species, 3.78% ± 1.18% for subspecies or semispecies, and 11.06% ± 0.53% for species within a genus. At mt COII, Collins and Dubach (2000), Ascunce et al. (2003), and Ruiz-García et al. (2014) reported, an average genetic distance of around 6% among species within a genus, and around 2–4% for subspecies. Bradley and Baker (2001) claimed, for mt Cyt-b, that values 11% would be indicative of specific recognition. Avise (1994) determined 5–7% of differences at the mt control region for different species and around 2% for subspecies in mammals. Additionally (and this was introduced in the new version of the manuscript), Tobe et al., (2010) showed that for mtCyt-b values higher than 2.5 % are considered enough to differentiate mammalian species (this was a suggestion of the Referee 3). Moreover, the genomes of the ocelot lineage is conservative and the genetic distances among species are usually small. The Hippotragus niger case should be more an exception than a generality.
L263: there is a discrepancy. Perhaps this should be phrased as: “.. we selected several specimens when available for each one …” Corrected
L379: please rephrase to: “Based on the dated BI-tree of complete mitogenomes…” Corrected
L379-383: the reader has no chance to assess this statement. Please provide a dated tree with a lime line at the bottom and the splits/knot you are referring to in the tree. The referee 1 claimed that the manuscript is too long and that many figures should be deleted. Henceforth, to introduce another figure could be no appropriate. The Bayesian trees exactly showed the same relationships between the Nariño cat and those most related taxa (Central America and Trans-Andean tigrina + (L. geoffroyi + L. guigna) at the mitogenomes, and L. colocola at the mtND5) that we showed in the ML trees (relationships among other clades could be different between Bayesian and ML trees). Thus, to enclose a new tree don’t apport new information about the relationships of the Nariño cat and the most related taxa and this would unnecessarily extend the manuscript.
L403: please rephrase (there is no such thing as “bootstrap percentages”). Perhaps: “lower bootstrap support” Corrected
L432: The BI does not do that. The Bayesian inference refers to the phylogenetic tree. Based on estimated posteriors for genetic distances a molecular clock can be applied (independent of the BI procedure) if a calibration date is given for one of the nodes. Corrected
L616-646: this can be substantially shortened L653-664: very lengthy speculations. This can be compressed to the statement that now the search is on for this species to be found in the wild. Corrected
Referee 1 said:
Figures:
Fig. 1. The pictures need to be larger and better focused. Fig. 1A should also be used in a figure panel (side by side) with the suppl. figs. 1-3 (see below) and Fig. 2. The reader can guess (from fig. 1) that the animal on the right is the narino cat (but this is not stated in the legend) The entire arrangement of the skin figures should be redone. The goal here should be to display the new species together (ideally side by side) with all those species that are the morphologically most similar ones. And then (potentially indicated by arrows in the pictures) the reader should be made aware of the differences that warrant species delineation. At the moment the reader has to move forth and back between pictures. For that purpose, it may help to remove all the “environmental pictures” (figs 3, 4, 5) and focus on comparative skin pictures as this would improve instructiveness for the reader. This suggestion is very important. Now, we have totally changes the Figures 1 to 5. All these figures, as well as the supplementary Figures 1-3, were edited in one unique Figure (now Figure 1). Now, the readers can simultaneously compare the aspect of the Nariño cat with all the other morphotypes considered. In Spanish we say: “An image is worth a thousand words”. Therefore, this new Figure 1, we believe is many more informative that the individual figures. Many thanks to Referee 1 for this suggestion.
Fig. 6: what’s missing here is a tree without the “new species”. The actual node, dividing colocolo from all others in that clade is not supported. To assess if this a problem in assigning species level to narino, one needs to know what the tree would look like without the new species. If the support for that node remains low and the other species in that clade are accepted, then this would remove any argument against assigning species level to narino. We don’t include the tree without the Nariño cat, but we did the ML tree without the Nariño cat and the tree was exactly the same to that with the Nariño cat and with similar bootstrap values in both trees.
Fig. 7: if kept, it should go to supplementals Fig. 8: if kept, should go to supplementals and should be visually improved Fig. 9: needs visual improvement Fig. 10: it provides no additional information (compared with Fig. 9), should be removed. As we previously mentioned, we kept Figure 7 (now Figure 3), Figure 8 (now Figure 4), and Figure 10 (now Figure 6) because they are important for different questions (we previously explained it). Additionally, we improved the quality of the figures (especially the names of the taxa in now Figures 2, 3, 5, and 6).
Supplemental figures 1-3: Why are they in the supplements? There should be a picture panel in the manuscript with the Narino cat pictures and these pictures side by side to allow direct comparison. Now these photos are in the new Figure 1.
Fig. S4 (which looks like it was superimposed on some former fig. 6) and Fig. S5 are of such low-resolution quality that I did not look at them. These supplementary figures were deleted.
Tables Table 1: What do empty “country” fields mean? Does it mean it is the same country as in the filed above? If so, than there shouldn’t be separating lines between these fields. Same is valid for the species (Don’t have separating lines between the species field if the species stays the same) Corrected
Table 2: if there is a chance to have that in portrait mode that would improve readability. Corrected
Table 3: same as for Table 2 Table 3 was deleted.
Table S1: Sequences are given but there are no references. As the tables need to be self-explanatory, references should be given here too. Corrected
Reference list: way too long Although it was difficult, some cites were deleted. Originally, there were 106 cites. Now, the manuscript has 98 cites.
Referee 2:
This referee did not really apport any comment to improve the quality of the manuscript. All his/her comments were negative and explaining everything that should be done in an ideal world research. However, many research is not done in an ideal and perfect world. Similarly, some of his/her statements do not fully match with the reality.
Referee 2 said:
This manuscript describes the molecular evidence for recognizing a skin of a small, spotted felid from southern Colombia as belonging to an undescribed species of the genus Leopardus. For context, the authors attempt to provide a molecular framework for understanding genetic relationships among members of the “ocelot” sub-lineage of the genus Leopardus, and they describe a new species based on seven variants in two mitochondrial genes and features of the pelage.
We did not describe a new species based on seven variants in two mitochondrial genes and features of the peleage. First, we detected a specimen (originally classified as L. tigrinus) with some differentiating features of the pelage and as a consequence of it, we investigated molecular traits of this skin (only the skin wouldn´t have been enough to define a new species). We did not define a new species based only in seven variants of two mitochondrial genes. We analyze complete mitogenomes (more than 16,000 base pairs) for 32 new specimens of the Leopardus genus. This is the major quantity of specimens of this genus analyzed for complete mitogenomes to date. Previously, only 12 complete mitogenomes of Leopardus specimens were analyzed. Additionally, we analyzed 56 new specimens of the Leopardus genera for the mtND5 gene and 113 new Leopardus specimens for six nuclear microsatellites. All these data showed us that this skin should belong to an undescribed Leopardus species and not the seven variants of two mitochondrial genes as the referee commented. Other different question is that, if some reader has samples of Leopardus and he/she wants to know if their samples could belong to the Nariño cat, we indicate the seven synapomorphies presented in the two genes more studied (major number of sequences in GenBank) to date in the Leopardus genus.
Referee 2 said:
Taxonomy and systematics of New World felids, although much improved, remains something of a mess, and one welcomes a manuscript that clearly defines a problem and utilizes competent scholarship and modern tools and resources to provide clear answers. Such clarity cannot be achieved, however, when “common” names and species and subspecies names are mixed and used interchangeably and inconsistently. This is a particular problem for felids where common names are often used as taxonomic names (the “margay” is Leopardus wiedii and Leopardus margay is a synonym of the “tigrina,” Leopardus tigrinus). The function of taxonomy is to provide an organism or grouping of organisms with a unique, formal Latin or latinized scientific name that cannot be confused with the names of other organisms or groups of organisms. The function of the rules in the International Code of Zoological Nomenclature and the rulings of the International Commission on Zoological Nomenclature is to provide stability of scientific names in zoology in order to prevent confusion. All of this effort is undermined when authors revert to common or vulgar names for which there no rules or standards and often are not commonly in use.
Two questions. First, the complexity and confusion of scientific names throughout the last centuries for the margay and the tigrina is not caused by us. Schreber (1775, 1777) was the first to use the scientific name of Felis tigrina and illustrated this species with a plate named “Le Margay” (Buffon 1765). It was based on an exemplar from Cayenne, French Guiana. Thus, since the beginning, biologists have been confused about the relationship between the tigrinas and the margays. Gray (1867) described a supposed new species, Felis pardinoides, with “India” as its type locality, but later changed the type locality to Bogota, Colombia (Gray 1874). In the same time period, Felis guttula was described from southern Brazil (Hensel 1872). Additional tigrina taxa were described in the beginning of the 20th century including F. p. oncilla (Costa Rica), F. p. andina (Ecuador), F. carrikeri (Costa Rica), F. p. emerita (Venezuela), and F. emiliae (type locality: Ipu, Ceara State, Brazil). Other author used the genus Margay and defined two additional holotypes of tigrina: M. t. elenae (Antioquia Department, Colombia) and M. caucensis (Cauca Department, Colombia) (Allen 1915). Later, the taxonomic confusion substantially increased between tigrinas and margays (Allen 1919). Two genera were differentiated, Margay and Oncilla with intermixed diverse taxa of small spotted cats in each. In Margay, two species were included. One was Margay tigrina with three subspecies: Margay tigrina tigrina (the original F. tigrina; Allen 1919: eastern Venezuela and northern Brazil; today L. tigrinus), Margay tigrina wiedii (southern Brazil, southwestern Colombia; today L. wiedii), and Margay tigrina vigens (Para, Brazil; today L. wiedii). The second species was Margay glaucula, which was comprised of two subspecies: Margay glaucula glaucula (Mexico; today L. wiedii) and Margay glaucula nicaraguae (Nicaragua; today L. wiedii). In Oncilla, three species were recognized. The first species Oncilla pardinoides had five subspecies: O. p. pardinoides, O. p. oncilla, O p. andina, O. p. emerita, and O. p. elenae. O. p. pardinoides inhabits Bogota, Colombia to southeastern Brazil. It is currently considered as L. tigrinus, although some specimens classified by other authors as Felis geoffroyi and Felis guigna belong to this taxon (Allen 1919). The second subspecies, O. p. oncilla inhabits Costa Rica. It contains the two holotypes of F. p. oncilla and F. carrikeri from Costa Rica (Allen 1919). Today it is referred to as L. tigrinus. The third subspecies, O. p. andina, was found in Ecuador and is now referred to as L. tigrinus. The four subspecies, O. p. emerita, is distributed in Merida, Venezuela and southward in the Eastern Colombian Andes to the Huila Department. Currently it is referred to as L. tigrinus. The fifth subspecies, O. p. elenae, inhabits Antioquia, Colombia, although it is similar to O. p. emerita. Today it is classified as L. tigrinus. The second species was Oncilla caucensis (Cauca, Colombia; today L. tigrinus). The third species was Oncilla guttula, with two subspecies: O. g. guttula (southern Brazil; today L. guttulus) and O. g. emiliae (Ceara, northeastern Brazil; today L. t. tigrinus or L. emiliae). The confusion increased in the succeeding decades. Cabrera and Yepes (1940) considered one unique species of tigrina classified in the genus Noctifelis. They identified it as Noctifelis pardinoides. However, others recognized a different species (Noctifelis guigna) in this genus (Allen 1919). In another approach, margay was recognized as Margay tigrina (Cabrera and Yepes 1940). Other author classified 10 subspecies of margay (current L. wiedii) within F. tigrinus (Weigel 1961): 1) L. t. tigrinus (from Para, Brazil, to Guyana), 2) L. tigrinus wiedii (southern Brazil and northern Argentina), 3) L. tigrinus boliviae (from Panama throughout the Colombian, Peruvian, and Bolivian Andes), 4) L. tigrinus amazonicus (Amazonas), 5) L. tigrinus nicaraguae (Honduras, Nicaragua, Costa Rica), 6) L. tigrinus salvinius (Belize, Honduras), 7) L. tigrinus yucatanicus (Yucatan and Chiapas, southern Mexico), 8) L. tigrinus glauculus (from Jalisco to Sinaloa, western Mexico), 9) L. tigrinus oaxacensis (Oaxaca and Veracruz, eastern Mexico), and 10) L. tigrinus cooperi (Texas, USA). The remaining forms of tigrina (excluding the original tigrina) were classified within the genus Oncifelis, O. pardinoides including four subspecies: O. pardinoides oncilla (Costa Rica), O. pardinoides pardinoides (western Venezuela, Colombia, Ecuador, and Peru), O. pardinoides emiliae (Ceara, Brazil), and O. pardinoides guttula (southern 696 Brazil) (Weigel 1961). The other two species of Oncifelis, were Oncifelis geoffroyi and Oncifelis guigna.
Therefore, we have not complicated the taxonomy of the tigrinas. Contrarily to that commented by Referee 2, we have tried to simplify the systematic and nomenclature of the tigrina to the maximum. Second. we used the term, Nariño cat, for brevity and for not introduce more difficulties in the reading of the manuscript. In the case of L. colocola, some authors use many sub-species and others many of these subspecies transformed them in species. We have adopted the most conservative point of view and cited them as subspecies. Therefore, this criticism seems unjustified.
Referee 2 said:
There are really two theses in this manuscript, neither of which is completely nor carefully realized. The first thesis is a series of molecular analyses of the “ocelot” sub-lineage of the genus Leopardus. The second idea is the delineation and description of a previously unrecognized species. Although the description derives from the molecular analyses, the molecular part of the study does not retain focus on the newly recognized lineage (actually this lineage was identified as early as Ruiz-García et al.2018), but often strays into issues of hybridization and relationships among other lineages. More importantly, it does not provide a clear answer regarding the relationships of the new taxon, providing multiple confusing alternative interpretations:
- “Nariño cat should be the most differentiated subspecies of L. colocola or a new sister species from L. colocola.” (lines 553–554)
- “The mitogenomic analysis showed the Nariño cat is the sister taxon of the clade formed by the Central American and trans-Andean tigrina + L. geoffroyi/L. guigna.” (554–556)
- “Based on the nuclear DNA microsatellite analysis, the Nariño cat is the sister taxon to the clade composed of L. geoffroyi + L. guigna.” (558–559)
- “Therefore, the Nariño cat should be a new subspecies of L. oncilla (this last taxon elevated to the category of species), or it is a new sister species of the Central American and trans-Andean tigrina (L. oncilla + (L. geoffroyi/L. guigna)).” (564–566)
Effectively, the manuscript wants to proof in two questions. First, to delineate a new taxon within the Leopardus genus. Second, to proof more on the systematic of the tigrinas (not for all the lineage of the Ocelot as claimed the Referee 2). Additionally, our molecular results identified two alternative hypothesis in relationship of the Nariño cat relationships within the Leopardus genus:
- The mtND5 gene (more specimens and more Leopardus taxa analyzed) identified the Nariño cat as the sister taxon from colocola.
- The complete mitogenomes and microsatellites (less specimens and a more restricted number of taxa) identified the Nariño cat as the sister taxon of Leopardus geoffroyi + Leopardus guigna. In the case of the complete mitogenomes in the sister taxon of the Nariño cat, it was also contained the Central American and Trans-Adean tigrinas. This last taxon was not analyzed for the microsatellites.
In whatever case, and this was not claimed by the Referee 2, the Nariño cat always appeared as a differentiated taxon or species from other taxa and species fully recognized within the Leopardus genus, independently of the taxa or species used, of the number of specimens used, the genetic markers used, and the different mathematical properties of the procedures used. This is that a new and deserves to be published.
Referee 2 said:
What is not clear is why the authors chose the analyses they did – what questions were they attempting to answer using each analysis? More important, why did they not follow up with additional analyses (and more nuclear genes or UCEs or …) to clarify the relationships?
It is clear why we used diverse molecular markers and different procedures. With the tools we have, from a Third World country (as Colombia is) where the economic resources are very limited, we did the maximum effort to sequence complete mitogenomes and then there were very few economic resources to analyze nuclear genes. For this reason, we could only analyze 6 nuclear microsatellites. In an ideal world, with unlimited economic resources, and many samples of the Nariño cat, and many, many samples of all the tigrinas described during centuries, we would surely have done another more complete research. But the real world is what it is. We did not have important economic resources (and with this limitation, yet, we generated the highest number of mitogenomes of the Leopardus genus to date), and only one sample of the Nariño cat (only a skin without other tissues) was found to date. This is the reality, which is not perfect.
Referee 2 said:
Another issue that plagues the molecular analysis is that it does not link genetic samples to the original voucher specimens. Based on the acknowledgments, the authors were able to obtain genetic samples from vouchers in a number of international collections, but the original vouchers are not listed by museum number anywhere in the paper, nor are all of the localities for the original specimens provided in GenBank records.
Referee 2 again suffered a mistake. With the exception of the Nariño cat sample, which was obtained in a museum, all the other samples (excluded the sequences from GenBank) proceeded from wild. During almost 20 years, the first author obtained all these samples from wild. Later we extracted DNA, we donated the tissues to the museum of our University. For this reason, there are not original vouchers from different international collections in any part of the manuscript. All the precise geographical origins of the samples that we obtained in the will, appeared in Table 1.
Referee 2 said:
The first question that needs to be addressed is whether the new lineage warrant a new name, or is there a previously proposed name that would apply to it? There are 12 synonyms of Leopardus tigrina (3 based on holotypes from Colombia) and myriad synonyms of other species of Leopardus that, based on the confused relationships of the new lineage, could apply to it. These need to be investigated thoroughly to make certain none apply
If the Referee 2 read the Introduction and the first part of the Results, that’s exactly we have done. We have tried to bring all the synonymies of the tigrinas into a number of acceptable taxa and compare them to the Nariño cat. From the pelage features certainly none applied to date, and molecularly speaking, we have analyzed the major number of tigrinas to date from different areas of Latin America, which tigrinas inhabit, and the Nariño cat is molecularly differentiated from all them. In fact, we detected four or five different probable tigrina species, being all them not monophyletic.
Referee 2 said:
The holotype of the new species is an extremely poor specimen, consisting of only a strangely stuffed skin. It lacks even a skull, which has provided the standard morphological characters used to delimit species for the past couple of centuries.
Again, in an ideal world the holotype of the Nariño cat could be an entire specimen, with skull, bones, skin, and other tissues. But the reality is that it is. All these differentiated molecular characteristics belonged to a unique dry skin !!!!! we cannot do anything more. Only to search for other skins or looking for some living specimen of this taxon (we are doing).
Referee 2 said:
While the authors do a passable job of describing the pelage pattern and colors, anyone who has worked with mammals knows that such characters differ among individuals and even within a single individual in different molts, and are inadequate to describe a new species without a comprehensive evaluation of intraspecific variation.
The Referee 2 again suffered a mistake. We are not defining the possible new species by the differentiated pelage characteristics (we are conscient that in some species there is an important variance for external characters within a same population. This is the basis for the synthetic Neodarwinism; Dobzhansky 1937, 1970; Mayr 1942, 1963, 1970, 2004, for instance). We are claiming for a new taxon or species of tigrina by the molecular results obtained.
Referee 2 said:
Using photographs of animals/skins is inadequate, as it is clear from Figs. 1–4 that each was taken under different lighting conditions.
We think that this comment is also from an ideal and perfect world. Obviously, the perfect situation should be to have all the skins of tigrinas of all the museums of the World, which have skins of this taxon in their collections and add them to this manuscript. But that, it’s not possible. It would also impossible to photograph them all together with the same conditions of light. This referee asks for things that cannot be executed in a real world. As we aforementioned “An image is worth a thousand words”. We believe that the pictures grouped in Figure 1 are better than whatever verbal and qualitative descriptions of the skins.
Referee 2 said:
Also an issue is fading of preserved skins. External measurements taken from a freshly killed specimen are of some minor value, as they vary with time after death (Stephens et al. 2015 J. Mamm. 96:185-193), but external measurement taken from a skin after it has been removed from a body are nearly worthless, as there is no means of knowing what kinds of stretching/shrinkage have taken place. One certainly cannot use the skin alone to determine roundness and width of the head, the flatness of the face, or the length or robustness of the body that is not there. While the authors claim to have made a comprehensive review of specimens in various collections, there is no list of specimens examined, which is a necessary part of a paper such as this intends to be
The question that in a dry skin, the measurements have not any value is possible. But we are not differentiated the Nariño cat as a new species by the measurements if its skin (our claim that it could be a new species is by the molecular results as we aforementioned) nor we are comparing these measurements with those of other tigrina taxa. Simply, we showed these measurements to the readers to have an approximate idea of the size of the specimen. That’s all. For this reason, an image is worth a thousand words.
Referee 2 said:
That the authors understand the inadequacy of their investigation is indicated in the last paragraph: “The following steps are needed to validate the existence of this new small spotted cat in the Neotropics. … locate additional skins and skulls similar to the Nariño cat in collections or museums. If other material is found, molecular and morphological analyses can assess the distinctiveness of and variation in the proposed taxon, lineage, or species.”
Once more time, this referee is wrong. This phrase of ours does not indicate that our research is inadequate, it indicates the limitations of this research and what should be done in the future to produce more complete results, which is a different concept.
Referee 2 said:
Instead of a follow-up study, however, this a necessary a priori step to their investigation. VertNet (http://vertnet.org/) lists 99 specimens of Leopardus tigrinus specimens in its database (which covers only a small portion of the world’s collections), 28 of which are from Colombia.
In an ideal world, we would have time and money to go to all museums in the world where they have those 99 specimens of tigrinas that the referee comments. Nevertheless, we are not in that ideal world, and we have neither the time nor the economic resources to visit all those museums. Indeed, we have observed the 28 samples of Colombian tigrinas in the museums of that country and no skin looks like that of the Nariño cat. Some of the skins that we showed in Figure 1 came from some of these specimens in Colombian museums.
Referee 2 said:
It is not clear that the authors understand the concept of a holotype
This comment is totally unnecessary.
Referee 3:
We thank very much to this referee because his/her observations were extremely useful to enhance the quality of the manuscript (correction on the manuscript in red).
Referee 3 said:
Extensive genetic data based on mtDNA and microsatellites appear to show that this specimen is distinct from all other known Leopardus species in Central and South America. The morphology of the skin appears also to be distinctive, but the authors have not considered that the reddish coloration may be due to an erythristic mutation.
This last sentence is very interesting (the reddish coloration may be due to an erythristic mutation), and we have added it to the new draft. However, such as it was previously commented, we considered the Nariño cat as a possible new species by the molecular genetics results and not by the characteristics of its pelage.
Referee 3 said:
The authors state that the results of mtDNA and microsats (nuclear DNA) are identical in establishing the distinctiveness of this specimen as a new taxon and they refute the possibility that this specimen could be some form of hybrid. The main issue with this paper is the sample size. It is difficult to be certain of the validity of a species based on one specimen especially in a genus where both ancient and current introgression is common. The authors state that microsatellites are not ideal for taxonomic studies and indeed it would be much better if SNPs could have been used.
We agree with Referee 3 that the existence of a unique specimen of the Nariño cat is the main problem. But this is the situation to date, and we have not found any other specimen with these phenotypical and molecular characteristics. However, we will look for. Also, we agree with this referee that other nuclear markers should better than microsatellites to undertake this task but with the economical resources that we had, it was what we could do. Nonetheless, we employed these microsatellites in other studies with felids and they showed a good signature to recover phylogenetic relationships inside different felid clades (for instance within the Panthera clade or with the Felis clade).
Referee 3 said:
The authors state that "Therefore, both the mitogenomic data and the nuclear microsatellite data offered identical results [lines 559-560]. However, reviewing Figures 6, 7 and 8, it seems to me that the placement of the Narino cat differs: In Fig. 6 it is sister to a clade that includes trans-Andean and Central American tigrinas plus Geoffroy's cat and guigna, in Figure 7 it is basal to a clade containing Leopardus colocola and hybrids, and in Figure 8 it is sister to Geoffroy's cat and guigna, so there are several dissimilarities in these relationships with other Leopardus taxa.
Such as it was previously commented (and it is commented in the new draft of the manuscript), complete mitogenomes and microsatellites both showed that the Nariño cat is the sister taxa of L. geoffroyi + L.guigna. The difference is that for complete mitogenomes, the Central American and Trans-Andean tigrinas were enclosed in the clade with L. geoffroyi + L.guigna, meanwhile these tigrinas were not analyzed with microsatellites. But the relationship of the Nariño cat with both L. geoffroyi + L.guigna is observable for both kinds of molecular markers.
Referee 3 said:
It would be good if the authors could also look at cytb. A study by Tobe et al. 2010 found that mammalian species differed by a K2P p-distance of 2.5 or greater.
It was a great idea of this referee. We did it, and some results are now included in the new draft. The results with mtCyt-b are very similar in their consequences to that obtained with mtND5 and the mitogenomes.
Referee 3 said:
The authors state that this species is extinct or close to extinction because no other specimens have turned up since its discovery in 1989. However, other small cat species appear to occur still in the collecting region and no suggestion was given as to why it may be or almost extinct. Perhaps publication of this new taxon may create the awareness to help discover new specimens to test the hypothesis raised here?
The unique specimen of Nariño cat was found in more than 3,000 meters on sea level in the paramo. This is an ecologically fragile area where the climatic change or whatever minimal anthropological pressure can affect to the species living there. The other cat species in the Nariño Department are living under 2,000 meters on sea level, where the ecosystems are not so fragile. On the other hand, the last sentence seems to be extremely important (“Perhaps publication of this new taxon may create the awareness to help discover new specimens to test the hypothesis raised here?”) and we have added it to the last paragraph of the manuscript.
Referee 3 said:
Tables 1 and 2 are really difficult to read. Simplification of some of the column and row labels would help enormously.
Table 1 has been simplified, and Table 2 was eliminated following the comments of Referee 1.
Referee 3 said:
I have made comments on the attached copy of this paper.
These corrections of English throughout all the text have been absolutely added to the new draft. We thank so much to this referee for this task that improved considerably the quality of the English.
Referee 3 said:
It would probably be better to use the two different dates independently because a mean of two with such a wide s.e or s.d (not clear) is probably not very robust (in reference to the molecular o’clock).
We also agree with this observation of Referee 3, but the Ocelot fossil record is extremely scarce and recent, and no valid dates are available. For this reason, we employed the two molecular dates more solid in the literature.
The authors considered that the comments of referees 1 and 3 help so much to improve the manuscript and that the comments of referee 2 are also important to get over some misunderstanding that a research, like this, may have. Thanks very much to all three.
Sincerely Yours,
Prof. Manuel Ruiz-García. PhD
Full Professor (Profesor Titular-Catedrático)
Coordinador Unidad de Genética
Departamento de Biología
Facultad de Ciencias
Pontificia Universidad Javeriana
Cra 7ª No 43-82
Bogotá-Colombia

Reviewer 2 Report
Morphological and genetic evidence for an undescribed Spotted Cat Species (Mammalia: Carnivora: Felidae: Leopardus) from the southern Colombian Andes
This manuscript describes the molecular evidence for recognizing a skin of a small, spotted felid from southern Colombia as belonging to an undescribed species of the genus Leopardus. For context, the authors attempt to provide a molecular framework for understanding genetic relationships among members of the “ocelot” sub-lineage of the genus Leopardus, and they describe a new species based on seven variants in two mitochondrial genes and features of the pelage.
Taxonomy and systematics of New World felids, although much improved, remains something of a mess, and one welcomes a manuscript that clearly defines a problem and utilizes competent scholarship and modern tools and resources to provide clear answers. Such clarity cannot be achieved, however, when “common” names and species and subspecies names are mixed and used interchangeably and inconsistently. This is a particular problem for felids where common names are often used as taxonomic names (the “margay” is Leopardus wiedii and Leopardus margay is a synonym of the “tigrina,” Leopardus tigrinus). The function of taxonomy is to provide an organism or grouping of organisms with a unique, formal Latin or latinized scientific name that cannot be confused with the names of other organisms or groups of organisms. The function of the rules in the International Code of Zoological Nomenclature and the rulings of the International Commission on Zoological Nomenclature is to provide stability of scientific names in zoology in order to prevent confusion. All of this effort is undermined when authors revert to common or vulgar names for which there no rules or standards and often are not commonly in use.
There are really two theses in this manuscript, neither of which is completely nor carefully realized. The first thesis is a series of molecular analyses of the “ocelot” sub-lineage of the genus Leopardus. The second idea is the delineation and description of a previously unrecognized species. Although the description derives from the molecular analyses, the molecular part of the study does not retain focus on the newly recognized lineage (actually this lineage was identified as early as Ruiz-García et al.2018), but often strays into issues of hybridization and relationships among other lineages. More importantly, it does not provide a clear answer regarding the relationships of the new taxon, providing multiple confusing alternative interpretations:
· “Nariño cat should be the most differentiated subspecies of L. colocola or a new sister species from L. colocola.” (lines 553–554)
· “The mitogenomic analysis showed the Nariño cat is the sister taxon of the clade formed by the Central American and trans-Andean tigrina + L. geoffroyi/L. guigna.” (554–556)
· “Based on the nuclear DNA microsatellite analysis, the Nariño cat is the sister taxon to the clade composed of L. geoffroyi + L. guigna.” (558–559)
· “Therefore, the Nariño cat should be a new subspecies of L. oncilla (this last taxon elevated to the category of species), or it is a new sister species of the Central American and trans-Andean tigrina (L. oncilla + (L. geoffroyi/L. guigna)).” (564–566)
What is not clear is why the authors chose the analyses they did – what questions were they attempting to answer using each analysis? More important, why did they not follow up with additional analyses (and more nuclear genes or UCEs or …) to clarify the relationships?
Another issue that plagues the molecular analysis is that it does not link genetic samples to the original voucher specimens. Based on the acknowledgments, the authors were able to obtain genetic samples from vouchers in a number of international collections, but the original vouchers are not listed by museum number anywhere in the paper, nor are all of the localities for the original specimens provided in GenBank records.
The second part of this manuscript dealing with the description of the newly recognized species simply does not meet the standards necessary to provide a new taxonomic name in the 21st century.
The first question that needs to be addressed is whether the new lineage warrant a new name, or is there a previously proposed name that would apply to it? There are 12 synonyms of Leopardus tigrina (3 based on holotypes from Colombia) and myriad synonyms of other species of Leopardus that, based on the confused relationships of the new lineage, could apply to it. These need to be investigated thoroughly to make certain none apply.
The holotype of the new species is an extremely poor specimen, consisting of only a strangely stuffed skin. It lacks even a skull, which has provided the standard morphological characters used to delimit species for the past couple of centuries. While the authors do a passable job of describing the pelage pattern and colors, anyone who has worked with mammals knows that such characters differ among individuals and even within a single individual in different molts, and are inadequate to describe a new species without a comprehensive evaluation of intraspecific variation. Using photographs of animals/skins is inadequate, as it is clear from Figs. 1–4 that each was taken under different lighting conditions. Also an issue is fading of preserved skins. External measurements taken from a freshly killed specimen are of some minor value, as they vary with time after death (Stephens et al. 2015 J. Mamm. 96:185-193), but external measurement taken from a skin after it has been removed from a body are nearly worthless, as there is no means of knowing what kinds of stretching/shrinkage have taken place. One certainly cannot use the skin alone to determine roundness and width of the head, the flatness of the face, or the length or robustness of the body that is not there. While the authors claim to have made a comprehensive review of specimens in various collections, there is no list of specimens examined, which is a necessary part of a paper such as this intends to be.
That the authors understand the inadequacy of their investigation is indicated in the last paragraph:
“The following steps are needed to validate the existence of this new small spotted cat in the Neotropics. … locate additional skins and skulls similar to the Nariño cat in collections or museums. If other material is found, molecular and morphological analyses can assess the distinctiveness of and variation in the proposed taxon, lineage, or species.”
Instead of a follow-up study, however, this a necessary a priori step to their investigation. VertNet (http://vertnet.org/) lists 99 specimens of Leopardus tigrinus specimens in its database (which covers only a small portion of the world’s collections), 28 of which are from Colombia.
Other issues:
It is not clear that the authors understand the concept of a holotype.
665-666: Supplementary Materials: The following supporting information can be downloaded at: www.mdpi.com/xxx/s1, Figure S1:
Going to this web site yields: Error 404 - File not found
Many of the sentences are in non-standard English. One of the authors is at an American university. Either he did not review this manuscript prior to submission, or ...
Author Response

(The authors gave the same response as above.)

Reviewer 3 Report
This is a revised version of this paper which I reviewed previously which makes the case for the recognition of a new felid species from Colombia based on a unique museum specimen. Extensive genetic data based on mtDNA and microsatellites appear to show that this specimen is distinct from all other known Leopardus species in Central and South America. The morphology of the skin appears also to be distinctive, but the authors have not considered that the reddish coloration may be due to an erythristic mutation. The authors state that the results of mtDNA and microsats (nuclear DNA) are identical in establishing the distinctiveness of this specimen as a new taxon and they refute the possibility that this specimen could be some form of hybrid.
The main issue with this paper is the sample size. It is difficult to be certain of the validity of a species based on one specimen especially in a genus where both ancient and current introgression is common. The authors state that microsatellites are not ideal for taxonomic studies and indeed it would be much better if SNPs could have been used.
The authors state that "Therefore, both the mitogenomic data and the nuclear microsatellite data offered identical results [lines 559-560]. However, reviewing Figures 6, 7 and 8, it seems to me that the placement of the Narino cat differs: In Fig. 6 it is sister to a clade that includes trans-Andean and Central American tigrinas plus Geoffroy's cat and guigna, in Figure 7 it is basal to a clade containing Leopardus colocola and hybrids, and in Figure 8 it is sister to Geoffroy's cat and guigna, so there are several dissimilarities in these relationships with other Leopardus taxa.
It would be good if the authors could also look at cytb. A study by Tobe et al. 2010 found that mammalian species differed by a K2P p-distance of 2.5 or greater.
Overall the authors have probably presented sufficient data to present a hypothesis for a new felid taxon (certainly South America would be a region where new felid taxa might be expected to turn up), but I am personally sceptical about this putative new taxon until more specimens turn up and feel that it is more likely to be an erythristic mutant of a known species (given that we have a less-than-complete knowledge of species boundaries and numbers in Leopardus). The authors state that this species is extinct or close to extinction because no other specimens have turned up since its discovery in 1989. However, other small cat species appear to occur still in the collecting region and no suggestion was given as to why it may be or almost extinct. Perhaps publication of this new taxon may create the awareness to help discover new specimens to test the hypothesis raised here?
Tables 1 and 2 are really difficult to read. Simplification of some of the column and row labels would help enormously.
I have made comments on the attached copy of this paper.

The English is mostly good but I have made some suggestions for improvements throughout
Author Response

(The authors gave the same response as above.)

Round 2
Reviewer 1 Report
General remarks:
The paper has been improved quite a lot from its previous version. However, several questions remain, which I outlined below.
Why the final tree presented was the ML tree, is not really clear to me. If the authors had used the BI tree, they could have included the ML Bootstrap support at the nodes anyway (together with the posterior probabilities), and they could have indicated the node age directly in the tree. The most critical node is the one that separates the OTUs from the second group from the top. With 42% bootstrap “support” it has essentially no support at all. It would have been interesting to know that the BI (PP) support was for that node.
Specific remarks
L26: odd sentence (sounds like the dating was performed 1.2-1.9 million years ago). Please change to “… was dated to 1.2-1.9 million years ago.”
L64: please remove the “by” after “comprised”
L145: closing bracket missing after “cruscinus”
L326: not sure, what means “were molecularly conformed”. Maybe the authors mean “were molecularly confirmed”?
L333/334: please change to “…nuclear microsatellite alleles)…”
L336: please change to ”…of the proposed new species, L. …”
ML tree (Fig. 2): as long as the “new species” has not yet been formerly recognized, throughout the paper one should use the terminus “proposed new species” instead of “new species”
367: please change to “numbers at the nodes indicate bootstrap support (%).”
L414: concerning Fig. 2. I don’t understand, why Fig. are so badly visualized. I cannot clearly read most of the node support values! But in addition, this tree is completely unresolved. Not only that, many of the sub-nodes provide no support at all?
Fig. 3: this figure should serve the purpose to scrutinize the relationship between the proposed new species and all other species that are in the same clade (L. colocola braccatus, Trans-Andean/Centro American trigrina, L. guigna, L. geoffroyi), in particular as the top branching within this clade has no bootstrap support (42%). In this context it is unclear, why F. catus and H.yagouaroundi are still used as outgroups (thereby compressing any distance among the Leopardus OTUs). If the ML tree is correct (as said above, I would have preferred to the BI tree) then using “northern Andean tigrina - and certainly without hybridization - as basal sister taxon to the clade above would serve the purpose much better. Such tree would also not require to include L. jacobita, L. wiedii, L. pardalis and “northern Andean tigrina hybridized with …”. The current ND5 tree (Fig. 3) is (as expected) way less informative than the ML tree before (Fig. 2).
L440-459: all percentages mentioned here fall from the sky as there is no table with these values. The only distance table provided by the authors is table 2 but that one deals with full mitogenomes and not with ND5 or cytb.
Fig. 4. Again, an unresolved tree, with poor intra-node support and apparently focussing on many intra-species relationships within the genus Leopardus. As the tree is shown how, I do not understand its purpose. From my view, the nDNA tree should be reconstructed to see if nDNA supports the distinctions made by the mtDNA tree. For that purpose, one would need the proposed new species, a L. colocola braccatus specimen (if no nDNA data exists for this one, then any other L. colocola subspecies would do), one Trans Andean/Centrao American tigrina specimen, and one L. guigna and one L. geoffroy specimen), and as outgroup a specimen from the northern Andean tigrina (w/o hybrid.).
All these numerous within species distances do not provide essential information for the reader for at least two reasons: 1) many of the species/subspecies appearing in this tree do not appear in the ML tree (Fig. 2), and 2) there is only a single specimen available for the proposed new species, so a within species variation cannot be assessed.
Fig. 5: This is an unrooted tree. Why does it contain H. yagouaroundi and F. catus? And again, the species names do not fully match the ones from the ML tree. This is very confusing for the reader (if the system of reference always changes).
Fig. 6: what additional purpose does this figure serve (compared with the tree Fig. 5)? There is no axis information, so that the spatial meaning is lost for the reader.
L519: please remove “by”
L584 onwards. I am not sure if this description is following the taxonomic code. If a new species is proposed, this has to follow the code (International Code of Zoological Nomenclature (ICZN)
aleady included in the comments above.
Author Response
SECOND REVIEW OF THE MANUSCRIPT ID: genes-2358502-Minor Revisions
Dear Editors of Genes,
We have revised and corrected the changes proposed by the referee (Minor Revisions). We thank to the referee their comments that helped to improve the quality of the paper. Herein are the changes made following to the referee:
- The referee said:
“The paper has been improved quite a lot from its previous version. However, several questions remain, which I outlined below.
Why the final tree presented was the ML tree, is not really clear to me. If the authors had used the BI tree, they could have included the ML Bootstrap support at the nodes anyway (together with the posterior probabilities), and they could have indicated the node age directly in the tree. The most critical node is the one that separates the OTUs from the second group from the top. With 42% bootstrap “support” it has essentially no support at all. It would have been interesting to know that the BI (PP) support was for that node.”
Now the Figure 2 includes the results for both the ML and BI trees such as suggested the referee. It is clear that the posterior probabilities (BI) are higher than the bootstraps (ML). The node with 42 % from bootstrap, with the posterior probabilities yielded 0.74. Thus, the change commented by the referee was enclosed in the new version of the manuscript.
- The referee said:
“L26: odd sentence (sounds like the dating was performed 1.2-1.9 million years ago). Please change to “… was dated to 1.2-1.9 million years ago.”
L64: please remove the “by” after “comprised”
L145: closing bracket missing after “cruscinus””
All these English corrections were done following to the referee.
- The referee said:
“L326: not sure, what means “were molecularly conformed”. Maybe the authors mean “were molecularly confirmed”?
L333/334: please change to “…nuclear microsatellite alleles)…”
L336: please change to ”…of the proposed new species, L. …””
All these changes of English and meaning have been done following to the referee.
- The referee said:
“ML tree (Fig. 2): as long as the “new species” has not yet been formerly recognized, throughout the paper one should use the terminus “proposed new species” instead of “new species””
We have corrected Figure 2 following to the referee. Now we write “proposed new species” instead of “new species”.
- The referee said:
“L367: please change to “numbers at the nodes indicate bootstrap support (%).””
It was corrected following to the referee.
- The referee said:
“L414: concerning Fig. 3. I don’t understand, why Fig. are so badly visualized. I cannot clearly read most of the node support values! But in addition, this tree is completely unresolved. Not only that, many of the sub-nodes provide no support at all?
Fig. 3: this figure should serve the purpose to scrutinize the relationship between the proposed new species and all other species that are in the same clade (L. colocola braccatus, Trans-Andean/Centro American trigrina, L. guigna, L. geoffroyi), in particular as the top branching within this clade has no bootstrap support (42%). In this context it is unclear, why F. catus and H.yagouaroundi are still used as outgroups (thereby compressing any distance among the Leopardus OTUs). If the ML tree is correct (as said above, I would have preferred to the BI tree) then using “northern Andean tigrina - and certainly without hybridization - as basal sister taxon to the clade above would serve the purpose much better. Such tree would also not require to include L. jacobita, L. wiedii, L. pardalis and “northern Andean tigrina hybridized with …”. The current ND5 tree (Fig. 3) is (as expected) way less informative than the ML tree before (Fig. 2).”
Because of the criticism of the referee in reference to this Figure 3 (we are partially in agreement with the referee), we deleted this figure.
- The referee said:
“L440-459: all percentages mentioned here fall from the sky as there is no table with these values. The only distance table provided by the authors is table 2 but that one deals with full mitogenomes and not with ND5 or cytb.”
We maintain Table 1 (genetic distances with mitogenomes) because it is a Table relatively small. However, the Tables of genetic distances with mtND5 and with mtCyt-b are very large and they occupied a lot of space. In the previous version of the manuscript, we enclosed the table with the genetic distances with mtND5 and all the referee suggested elimination because it was very large. Therefore, we eliminated the tables with these genetic distances. However, we have maintained in the text some genetic distance values because they are important to show as these values between the Nariño cat and other Leopardus taxa are of an elevated magnitude. In fact, as cited in the text, the genetic distance values with these mitochondrial genes are frequently used to determine different mammalian species, in general, and carnivore species in particular. It is a proof in favor that the Nariño cat is a different species from other Leopardus taxa.
- The referee said:
“Fig. 4. Again, an unresolved tree, with poor intra-node support and apparently focussing on many intra-species relationships within the genus Leopardus. As the tree is shown how, I do not understand its purpose. From my view, the nDNA tree should be reconstructed to see if nDNA supports the distinctions made by the mtDNA tree. For that purpose, one would need the proposed new species, a L. colocola braccatus specimen (if no nDNA data exists for this one, then any other L. colocola subspecies would do), one Trans Andean/Centrao American tigrina specimen, and one L. guigna and one L. geoffroy specimen), and as outgroup a specimen from the northern Andean tigrina (w/o hybrid.).
All these numerous within species distances do not provide essential information for the reader for at least two reasons: 1) many of the species/subspecies appearing in this tree do not appear in the ML tree (Fig. 2), and 2) there is only a single specimen available for the proposed new species, so a within species variation cannot be assessed.”
For us, this tree with the nuclear microsatellite data set is important. First, this is a tree done with specimens (not with species as unique OTUs as the two following figures; It is important to see as changing the number of OTUs, the phylogenetic relationships of the Nariño cat did not change). It is certain that the bootstrap percentages are low in this tree, but still the Nariño cat is always the sister taxon of L. geoffroyi/L. guigna as it was shown in the mitogenomic tree. On the other hand, the referee should understand that in this work we worked with 3 data sets (as they are explained in Methods): one data set for mitogenomes, one data set for mtND5 gen, and one data set for nuclear microsatellites. And each of these data sets have different Leopardus taxa and different number of specimens. For example, the mitogenomic data set included many more specimens of tigrinas (pardinoides and not pardinoides), of L. guttulus, of Central American and trans-Andean tigrinas that (unfortunately) the microsatellite data set. On the contrary, the microsatellite data set included subspecies of L. colocola that we (unfortunately) did not analyze for complete mitogenomes (these subspecies were also analyzed for mtND5). This was due to the fact that some samples were not enough to obtain DNA for both procedures, and these procedures were carried out in different years. For this reason, we maintained this figure in the new version of the manuscript.
- The referee said:
“Fig. 5: This is an unrooted tree. Why does it contain H. yagouaroundi and F. catus? And again, the species names do not fully match the ones from the ML tree. This is very confusing for the reader (if the system of reference always changes).”
As claimed the referee, we eliminated H. yagouaroundi and F. catus of this analysis, although the result was the same. The referee again commented that “...the species names do not fully match the ones from the ML tree…”. Obviously, they did not match because each one of these analyses were carried out with different data sets that they did not enclose the same species and the same subspecies as we previously said.
- The referee said:
“Fig. 6: what additional purpose does this figure serve (compared with the tree Fig. 5)? There is no axis information, so that the spatial meaning is lost for the reader.”
We maintained this Figure (now Figure 5) because, we believe that it is important. It is important to show that the relationships between the Nariño cat and L. geoffroyi/L. guigna is not caused by one particular procedure with some mathematical properties. The use of very different mathematical procedures (as an unrooted unweighted neighbour-joining procedure with Euclidian distances and as Principal Coordinate Analysis with the Gower’s procedure and the dm2 genetic distance) is indifferent to the strong relationship between the Nariño cat and L. geoffroyi and L. guigna, two species far away 4,000 km from southern Colombia where the Nariño cat was sampled. To show this, is very important.
- The referee said:
“L519: please remove “by””
It was corrected following to the referee.
- The referee said:
“I am not sure if this description is following the taxonomic code. If a new species is proposed, this has to follow the code (International Code of Zoological Nomenclature (ICZN)”
We followed the rules of the International Code of Zoological Nomenclature.
All the English corrections suggested by this referee were added in this new version of the manuscript.
Thanks so much, for the time and the comments carried out by the referees to improve the quality of the manuscript.
Sincerely Yours,
Prof. Manuel Ruiz-García. PhD
Full Professor (Profesor Titular-Catedrático)
Coordinador Unidad de Genética
Departamento de Biología
Facultad de Ciencias
Pontificia Universidad Javeriana
Cra 7ª No 43-82
Bogotá-Colombia

Reviewer 3 Report
This version of the paper is much improved and there are only minor edits that the authors might consider depending on the journal style.
I remain concerned about the description of a new species based on a single specimen but the authors' genetic data appear to show that this could be a new taxon. It is hard to imagine why this new species might be now extinct given that other small cats persist in the region and also what ecological niche it occupies that is different to these species. I think it is fair enough that this is presented as a hypothesis which can be tested when and if further specimens become available.

The quality of the English is good with only minor edits required.
Author Response
Dear Editors of Genes,
After the minor corrections of referee 1, I made now the minor corrections of referee 3. In the current manuscript draft I send you, I enclosed the corrections of both referees (they could be seen in red).
All the English corrections carried out by referee 3 have been enclosed in the current manuscript. Additionally, the quality of the names of the current Figure 3 is better.
Therefore, the major part of the suggestions of referee 1 and all the suggestions of referee 3 were added to the manuscript that I send you.
Thanks so much to both referee by their important suggestions which improved the quality of the article.
Sincerely Yours
Prof. Manuel Ruiz-García. PhD
Full Professor (Catedrático)
Coordinator Genetic Unit
Departamento de Biología
Facultad de Ciencias
Pontificia Universidad Javeriana
Cra 7ª No 43-82
Bogotá DC, Colombia
AND
Associate Research of the
Instituto Nacional de Biodiversidad del Ecuador
(INABIO), Quito, Ecuador.
